# Precise spatial tuning of visually driven alpha oscillations in human visual cortex

Kenichi Yuasa[1,2]*, Iris IA Groen[1,3], Giovanni Piantoni[4], Stephanie Montenegro[5], Adeen Flinker[5], Sasha Devore[5], Orrin Devinsky[5], Werner Doyle[5], Patricia Dugan[5], Daniel Friedman[5], Nick F Ramsey[4], Natalia Petridou[4], Jonathan Winawer[1,6]*

[1]Department of Psychology, New York University, New York, United States; [2]Division of Neural Dynamics, Department of System Neuroscience, National Institute for Physiological Sciences, Okazaki, Japan; [3]Video & Image Sense Lab, Informatics Institute, University of Amsterdam, Amsterdam, Netherlands; [4]University Medical Center Utrecht, Utrecht, Netherlands; [5]New York University School of Medicine, New York, United States; [6]Center for Neural Science, New York University, New York, United States

**\*For correspondence:**
kenyuasa@nips.ac.jp (KY);
jonathan.winawer@nyu.edu (JW)

## eLife Assessment

This intracranial EEG study presents **important** and **convincing** neural evidence supporting the high spatial specificity (receptive field) of visually driven alpha-band oscillation in human brains and its potential role in exogenous cuing attention. The work challenges the predominant view about the role of alpha-band oscillation in visual attention and advocates that stimulus-driven alpha suppression is precisely tuned and might contribute to exogenous spatial attention.

**Abstract** Neuronal oscillations at about 10 Hz, called alpha oscillations, are often thought to arise from synchronous activity across the occipital cortex and are usually largest when the cortex is inactive. However, recent studies measuring visual receptive fields have reported that local alpha power increases when cortex is excited by visual stimulation. This contrasts with the expectation that alpha oscillations are associated with cortical inactivity. Here, we used intracranial electrodes in human patients to measure alpha oscillations in response to visual stimuli whose location varied systematically across the visual field. We hypothesized that stimulus-driven local increases in alpha power result from a mixture of two effects: a reduction in alpha oscillatory power and a simultaneous increase in broadband power. To test this, we implemented a model to separate these components. The two components were then independently fit by population receptive field (pRF) models. We find that the alpha pRFs have similar center locations to pRFs estimated from broadband power but are several times larger and exhibit the opposite effect: alpha oscillatory power decreases in response to stimuli within the receptive field, reinforcing the link between alpha oscillations and cortical inactivity, whereas broadband power increases. The results demonstrate that alpha suppression in the human visual cortex can be precisely tuned, but that to measure these effects, it is essential to separate the oscillatory signal from broadband power changes. Finally, we show how the large size and the negative valence of alpha pRFs can explain key features of exogenous visual attention.

## Introduction

Hans Berger first measured electrical oscillations at 10 Hz from the human brain using EEG (*Berger, 1929*). Shortly thereafter, *Adrian and Matthews, 1934* confirmed that when a participant closes their eyes or views a uniform field, the oscillatory power increases. They described the alpha oscillation as

'show[ing] the negative rather than the positive side of cerebral activity; it shows what happens in an area of cortex which has nothing to do, and it disappears as soon as the area resumes its normal work.' Consistent with this interpretation, alpha oscillatory power is reduced by arousal (*Barry et al., 2007*) and alertness (*Matousek and Petersén, 1983*). The negative association between alpha oscillations and cerebral activity can also be more specific. For example, when attending a location to the left or right of fixation, the occipital alpha power measured by EEG decreases contralaterally and increases ipsilaterally (*Kelly et al., 2006*; *Sauseng et al., 2005*; *Worden et al., 2000*), opposite to the fMRI response (reviewed by *Carrasco, 2011*; *Somers et al., 1999*). The reduction in alpha power can also be specific to cortical locations representing narrow angular wedges in the visual field, as measured by both EEG and MEG (*Foster et al., 2017*; *Popov et al., 2019*; *Rihs et al., 2007*; *Samaha et al., 2016*). The common finding across these studies is that the strength of the alpha rhythm is lower in the part of cortex that is more active.

It is, therefore, surprising that studies examining the spatial tuning of alpha oscillations with invasive methods have tended to find a positive association between alpha oscillations and cortical activity. Specifically, two studies in macaque measured spatial receptive fields using power in the alpha band as a dependent measure, and found that for most recording sites, alpha power increased rather than decreased when a stimulus was in the receptive field (*Klink et al., 2021*; *Takaura et al., 2016*). A positive association has also been found in human intracortical studies, reporting power increases in the alpha band when stimuli were in or near electrode receptive fields (*Harvey et al., 2013*; *Luo et al., 2024*).

Why do the intracranial studies on spatial tuning find increases in alpha oscillations, whereas the MEG and EEG studies find decreases? One difference is that cortical engagement was stimulus-driven in the intracranial studies and task-driven (spatial attention) in the MEG and EEG studies. We speculate that both visual stimulation and visual attention tend to decrease alpha oscillations, but that visual stimulation masks the alpha decrease with a simultaneous increase in broadband neural responses. Broadband spectral power increases are observed in many intracranial studies in association with increased neural activity (*Crone et al., 1998*; *Hermes et al., 2017*; *Miller et al., 2009a*; *Winawer et al., 2013*) and likely have a distinct biological cause from oscillations (*Hermes et al., 2017*; *Ray and Maunsell, 2011*). Because a visual stimulus can cause a simultaneous increase in broadband power and decrease in alpha oscillatory power, a power change at the alpha frequency is ambiguous. Separating responses by frequency band does not resolve the ambiguity because broadband power changes can extend to low frequencies, including the alpha band (*Winawer et al., 2013*).

Here, we examined how alpha oscillations are modulated in human visual cortex by visual stimulation using electrocorticographic (ECoG) recordings in nine participants. We addressed two main questions: First, are alpha oscillations increased or decreased by visual stimulation? Second, what is the spatial specificity of the modulation of alpha oscillations? To answer these questions, we used a modeling approach to separate the alpha oscillation from non-oscillatory (broadband) responses. We then used pRF modeling to quantify the spatial tuning of separate components of the ECoG signal, corresponding to alpha oscillations and broadband responses.

By separating the ECoG voltage time course into distinct components, we find that the broadband signal is increased and the alpha oscillation is decreased by visual stimulation. We also find that the broadband pRFs and alpha pRFs have similar locations (center positions) but differ in size. Together, the results suggest that the alpha oscillation in human visual cortex is spatially tuned and negatively modulated by visual input. In the Discussion, we consider how the spatially tuned reduction in the alpha oscillation may contribute to visual function, showing a surprisingly tight link between the spatial profiles of the stimulus-driven alpha oscillation on the one hand, and the behavioral effects of spatial attention on the other hand. Finally, we show how the former may give rise to the latter.

## Results
### Two signatures of visual responses measured with electrocorticography
Visual stimulation causes a variety of temporal response patterns measured in visual cortex. In addition to the evoked potential, which is a characteristic change in voltage time-locked to the stimulus onset, there are also spectral perturbations (*Makeig, 1993*). The largest such perturbation is the alpha oscillation. Typically, when the eyes are closed or when the participant views a blank screen (no

stimulus contrast), the field potential measured from visual cortex oscillates at about 8–13 Hz (*Berger, 1929*). Opening the eyes or viewing a stimulus reduces this oscillation. In addition to these spectral perturbations, there is also a broadband response, an increase in power across a wide frequency range following stimulus onset (*Manning et al., 2009*; *Miller et al., 2009a*). Both the broadband increase and alpha suppression can be elicited by the same stimulus and measured from the same electrode. In the example shown in *Figure 1*, stimulus onset caused a broadband increase in power spanning the full range plotted, from 1 to 150 Hz, and it eliminated the peak in the alpha range. The two types of response cannot be separated by simply applying filters with different temporal frequency ranges because the signals overlap in the low frequency range (6–22 Hz in *Figure 1f*).

## Separating the alpha oscillation from broadband power

Because stimulus onset tends to cause an increase in broadband power and a decrease in the alpha oscillation, it is possible for the two effects to cancel at some frequencies. In the example shown in *Figure 1*, the power at the peak of the alpha oscillation (~13 Hz, black dashed line) is nearly the same during stimulus and blank. It would be incorrect, however, to infer that the alpha oscillation was unaffected by the stimulus. Rather, the decrease in the alpha oscillation and the increase in broadband power were about equal in magnitude. Depending on the relative size of these two spectral responses, the power at the alpha frequency can increase, decrease, or not change. We illustrate these possibilities with three nearby electrodes from one patient (*Figure 2*). For all three electrodes as shown in the left panels, there is a peak in the alpha range during blanks (black lines) but not during visual stimulation (brown lines). Due to the interaction with the broadband response, the power at the peak alpha frequency for stimulus relative to blank increased slightly for one electrode (Oc18), decreased slightly for a second electrode (Oc17), and decreased more substantially for a third electrode (Oc25). By adjusting the baseline to account for the broadband response as shown in the right panels, it is clear that the stimulus caused the alpha oscillation to decrease for all three electrodes. For these reasons, simply comparing the spectral power at the alpha frequency between two experimental conditions is not a good indicator of the strength of the alpha rhythm. To separate the alpha oscillation from the broadband change, we applied a baseline correction using a model-based approach.

## Alpha responses are accurately predicted by a population receptive field model

By estimating the alpha oscillatory power change from the low-frequency decomposition and the broadband power change across the high frequency range for each stimulus location, we obtain two time-courses per electrode, one for the alpha suppression (*Equations 1; 2*) and one for the broadband elevation (*Equation 3*; *Figure 3*). Each point in these time courses is a summary measure of a response component for a 500 ms stimulus presentation. Hence, the time series are at very different scales from the ms voltage time series used to derive the summary measures.

After a small amount of pre-processing (averaging across repeated runs, temporal decimation, dividing out the mean response during the blanks), we separately fit pRF models to the broadband and the alpha time series (*Figure 3*). The pRF model we fit assumes linear spatial summation, with a circularly symmetric Difference of Gaussians (but constrained so that the surrounding Gaussian is large compared to the field of view of the experiment). Similar results were obtained using a model with compressive spatial summation or a model in which the surround size was allowed to vary.

In the example time series, it is evident that both components of the response vary with the stimulus position: for certain stimulus locations, the broadband response increased about 5- to 10-fold over baseline, and the alpha response decreased about 3- to 5-fold. The pRF models accurately predicted these time series, with twofold cross-validated variance explained of 90.2% for broadband and 69.5% for alpha. For both types of pRF, the center locations are in the parafovea in the left lower visual field, but the alpha pRF is about twice the size of the broadband pRF (*Figure 3c*).

Next, we quantified the prediction accuracy of the pRF model fit across many electrodes, participants, and visual field maps. We first selected a large set of electrodes based only on location (see *Electrode Localization* in Methods 4.6). Electrodes were assigned to visual areas probabilistically. This implies that some electrodes were assigned to multiple visual areas, consistent with our uncertainty about localization. Probabilistic assignment results in high probability assignments getting more weight. We then grouped the electrodes based on whether or not we expected them to be visually

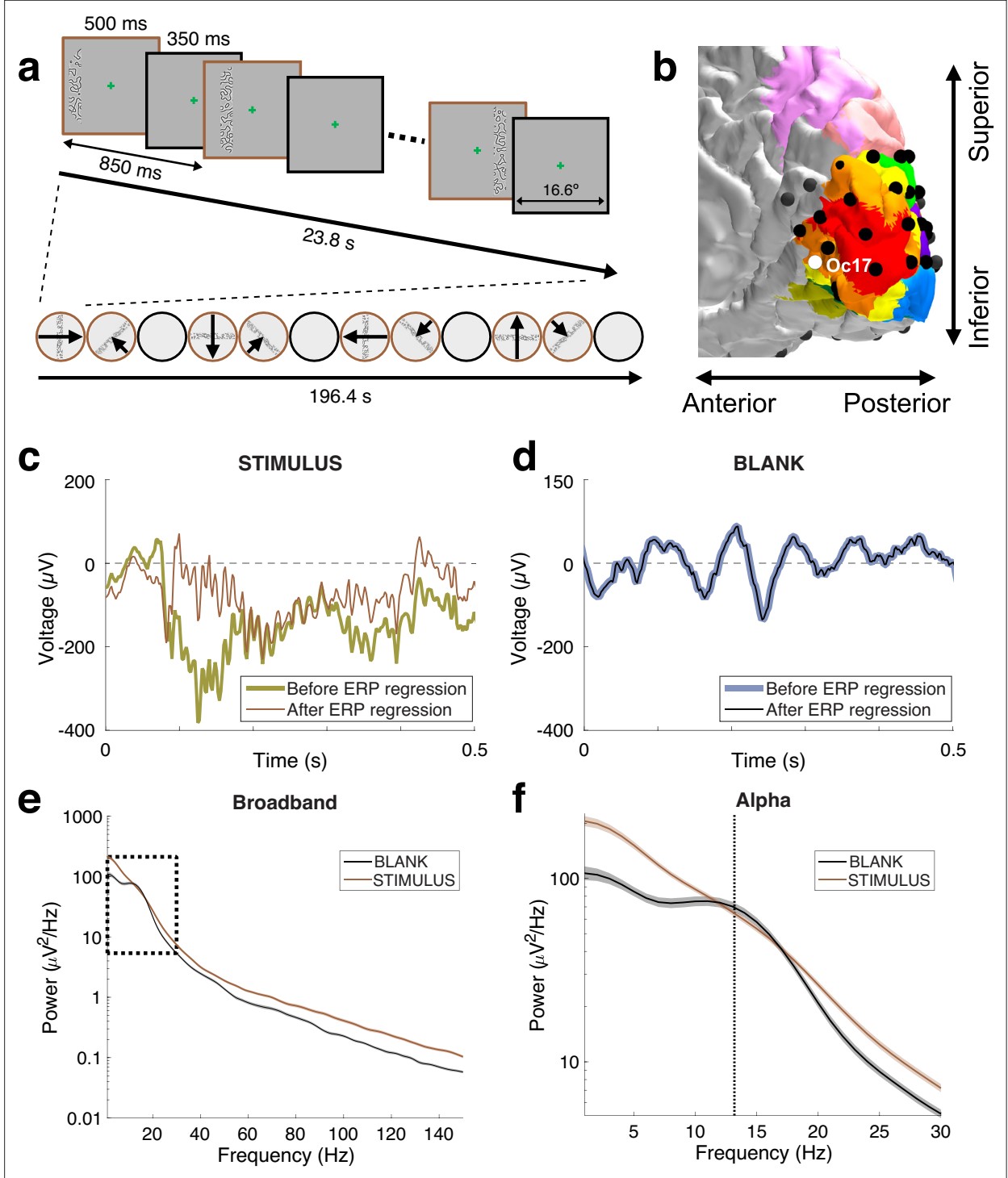

**Figure 1.** Two signatures of visual responses measured with electrocorticography (ECoG). (**a**) Mapping stimuli. Three examples of the bar stimuli and one image of the blank stimulus during the mapping experiment (brown and black outlines, respectively). The contrast within the bar is increased here for visibility. (**b**) A close-up view of the right occipital cortex of Patient 2. The surface colors show several visual field maps derived from an atlas, with V1 in red, V2 in orange, and V3 in yellow. Black circles indicate ECoG electrode locations. (See *Figure 1—figure supplement 1* for all patients) (**c**) The voltage signal during a single trial with a stimulus present for electrode Oc17. The thick and thin lines are voltages before and after regressing out the ERP, respectively. Note the sharp high frequency oscillations in both traces, and the lack of a clear alpha oscillation. (**d**) Same but for a single blank trial. Here, the two signals are nearly identical since there is no ERP during a blank trial. Note the strong alpha oscillation. (See *Figure 1—figure supplement 2* for the effect of ERP removal processing in population receptive field, pRF estimation.) (**e**) The power spectral density (PSD) was computed based on the regressed signal, averaged across all blank trials (black) or all stimulus trials (brown). The stimulus causes a broadband power increase, spanning

*Figure 1 continued on next page*

*Figure 1 continued*

frequencies from 1 to 150 Hz. The dotted black rectangle indicates the frequency range shown in zoom in panel (**f**). (**f**) During the blank, but not during visual stimulation, there is a peak in the spectrum at around 13 Hz (alpha oscillation, dashed vertical line). Shading represents the standard error across trials (320 trials for stimulus, 128 trials for blank). See makeFigure1.m.

The online version of this article includes the following figure supplement(s) for figure 1:

**Figure supplement 1.** Electrode locations and visual areas.

**Figure supplement 2.** Population receptive field (pRF) model comparisons with and without ERP-Regression Removal.

responsive. Some of these electrodes are not expected to be visually responsive, for example, if their receptive field is beyond the stimulus extent. We identified visually responsive electrodes as those whose broadband pRF model accurately predicted the broadband time course (*Figure 4—figure supplement 1*, left). Because the broadband quantification is limited to high frequencies (70–180 Hz), this criterion provides an independent means to separate the electrodes into groups to assess the alpha pRF model accuracy, which only depends on signals below 30 Hz.

We find that for visually selective electrodes in V1 to V3, the alpha pRF model explains 37% of the cross-validated variance in the alpha time series (*Figure 4*, 'All Patients'). The non-visually selective

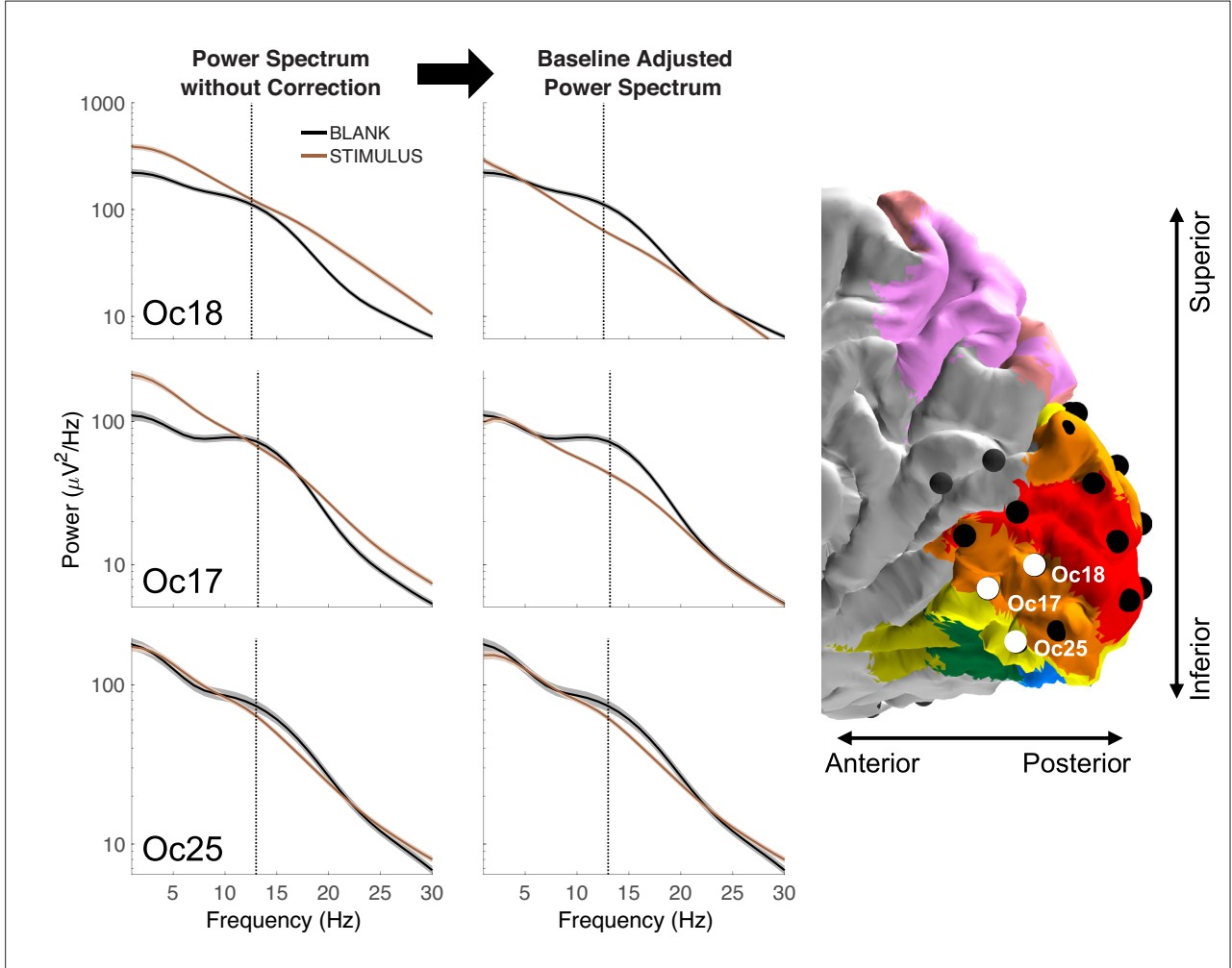

**Figure 2.** Interaction between the alpha rhythm and broadband power. (Left column) power spectral densities (PSDs) from three electrodes in Patient 2 without baseline correction show that the stimulus-related alpha power can slightly increase (Oc18), slightly decrease (Oc17), or substantially decrease (Oc25) relative to a blank stimulus. (Right column) After correcting for the broadband shift, it is clear that for all three electrodes, the alpha oscillation is suppressed by visual stimulation. The dashed vertical lines indicate the alpha peak frequency. Shading represents the standard error across trials (320 trials for stimulus, 128 trials for blank). See Methods and makeFigure2.m.

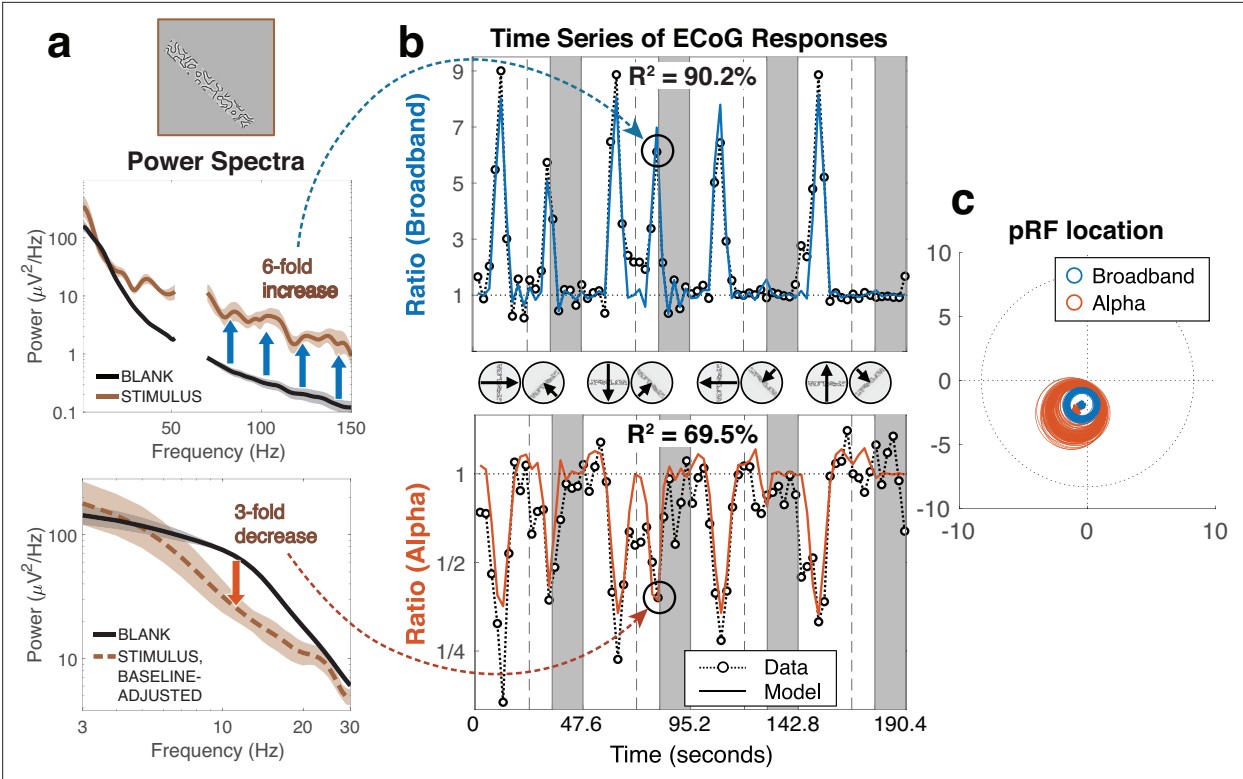

**Figure 3.** Example time series and population receptive field (pRF) fits. Representative data from a V3 electrode ('GB103') from Patient 8. (**a**) The computation of broadband and alpha summary metrics for an example stimulus location. The power spectra were computed over 500 ms of stimulus presentations or blanks. The dashed brown line in the lower panel indicates spectral power for the stimulus after correcting for broadband shift. Shading represents the 68% bootstrapped confidence intervals across the 6 runs for STIMULUS and across 384 trials for BLANK, each with 1000 resamples. (**b**) The time series of the broadband and alpha summary metrics averaged across six 190 s mapping runs, during which bar stimuli swept the visual field 8 times (with four blank periods indicated by the gray bars). Each data point (black circle) is a summary value—either broadband or alpha—for one stimulus position (*Equations 1–3*). Predicted responses from a pRF model are shown as solid lines. Model predictions below baseline in broadband and above baseline in alpha were possible because of the Difference of Gaussians (DoG) pRF model (see *Figure 3—figure supplement 1*). (**c**) The location of the pRFs fitted to the broadband and alpha times series. The circles show the 1-sigma pRF bootstrapped 100 times across the six repeated runs. See makeFigure3.m.

The online version of this article includes the following figure supplement(s) for figure 3:

**Figure supplement 1.** Population receptive field (pRF) model comparisons between two Gaussian models.

electrodes provide a null distribution for comparison, and for these electrodes, the pRF model explains close to 0% of the variance. This pattern holds for individual patients with electrodes in V1–V3 (*Figure 4*, 'Individual Patients'). In dorsolateral maps beyond V1–V3, the alpha pRF model explained about 24% of the variance in visually responsive electrodes, and no variance in the non-responsive group (*Figure 4—figure supplement 2*).

Overall, the difference in variance explained for the visually responsive vs non-visually response electrodes is many times larger than the variability across electrodes. These results show that across visual cortex, the pRF models explain a substantial part of the variance in the alpha time series.

## The alpha pRFs are larger than broadband pRFs, but have similar locations

We next compared the alpha and broadband pRF parameters. We limited these comparisons to electrodes for which both signal types were well fit by pRF models (*Figure 4—figure supplement 1*) and whose pRF centers were within the maximal stimulus extent (8.3°) for both broadband and alpha. Fifty-three electrodes met these criteria, of which 35 are assigned to V1–V3 and 45 to dorsolateral maps. (Because the assignments are probabilistic, 27 electrodes are assigned to both visual areas.)

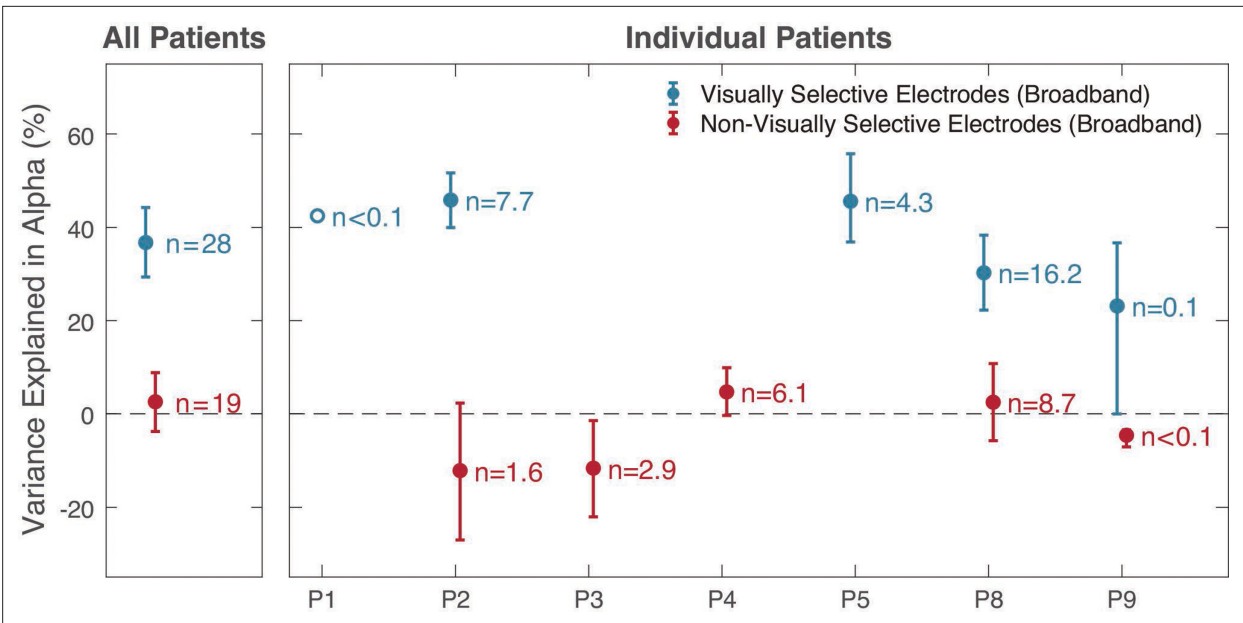

**Figure 4.** Prediction accuracy of alpha population receptive field (pRF) models in V1–V3. The variance explained by alpha pRF models was higher in visually selective than non-visually selective electrodes. The left panel shows the results pooled across patients. The right side shows results from each patient separately. Two patients who didn't have electrodes in V1–V3 are not displayed. Visually selective electrodes were defined as those whose broadband pRF model accurately predicted the broadband time course (see *Figure 4—figure supplement 1*). A similar pattern was observed for Dorsolateral maps beyond V1–V3 (*Figure 4—figure supplement 2*). Electrodes were assigned to visual field maps probabilistically across 5000 samples. Data points and error bars represent means ± 1 standard deviation of the mean across the 5000 samples. An open dot includes only one electrode. See makeFigure4_4S1_4S2.m.

The online version of this article includes the following figure supplement(s) for figure 4:

**Figure supplement 1.** Distributions of variance explained in alpha and broadband population receptive fields (pRFs).

**Figure supplement 2.** Prediction accuracy of alpha population receptive field (pRF) model in dorsolateral.

As with the sample electrode (*Figure 3*), the center locations of the two pRF types tend to be close (*Figure 5a*). However, the alpha pRFs tend to be much larger and slightly more peripheral. The larger size of alpha pRFs is found in each patient and in nearly every electrode (15 of 17 for V1–V3, 34 of 36 for dorsolateral electrodes, associated with maximum probability; *Figure 5a*, *Figure 5—figure supplement 1a*). Because the alpha pRFs are larger and the center locations are similar, the broadband pRFs are almost always contained within the alpha pRFs. Truncating the pRF sizes at 1 standard deviation, the percentage of the broadband pRF inside the alpha pRF was 92.3% (V1–V3) and 98.0% (dorsolateral) pRFs. This overlap is not a trivial consequence of the alpha pRFs simply being large. If we shuffle the relationship between alpha and broadband pRFs across electrodes, the overlap decreases sharply, to 30.0% (V1–V3) and 25.7% (dorsolateral). We summarized the relationship between the broadband and alpha pRFs by normalizing the parameters into a common space (*Figure 5b*, *Figure 5—figure supplement 1b*) and then averaging pRFs across electrodes for both alpha and broadband. This analysis again confirms that the broadband pRF is, on average, smaller and less peripheral than the alpha pRF, and is mostly inside the alpha pRF.

In addition, a direct comparison of the alpha and broadband pRFs for each separate pRF model parameter (polar angle, eccentricity, size) reveals several patterns (*Figure 5c*, *Figure 5—figure supplement 1c*). First, the polar angles are highly similar between broadband and alpha both in V1–V3 ($r=0.98$) and dorsolateral visual areas ($r=0.95$). Second, the eccentricities are correlated but not equal ($r=0.68, 0.09$): the alpha pRFs tend to have larger eccentricities than the broadband pRFs, especially in dorsolateral visual areas. Third, the sizes of alpha pRFs also correlate with those of broadband pRFs ($r=0.33, 0.24$), but are about 2–4 times larger. These results suggest that the same retinotopic maps underlie both the alpha suppression and the broadband elevation, but that the alpha suppression pools over a larger extent of visual space. In the Discussion, we speculate about why the eccentricity

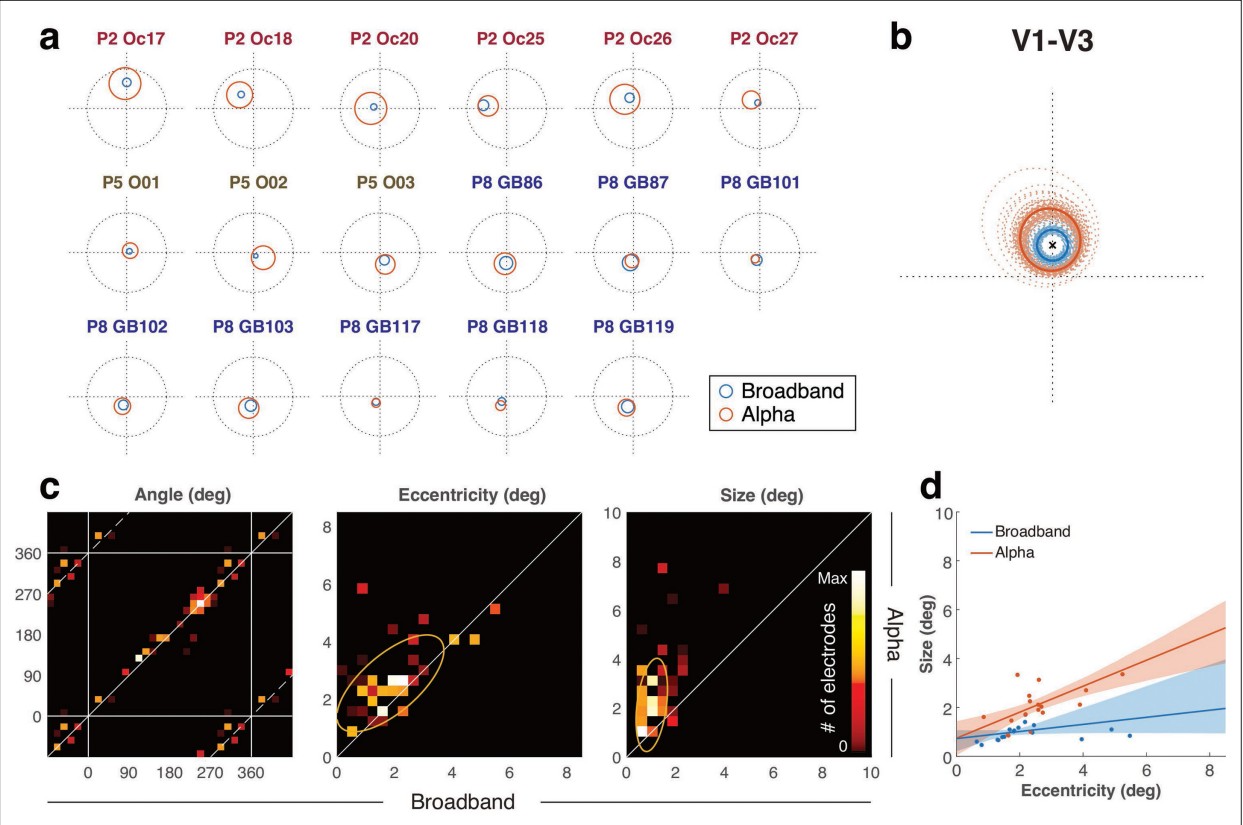

**Figure 5.** Relationship between alpha and broadband population receptive fields (pRFs) in V1–V3. (**a**) pRF locations in V1–V3 displayed for each electrode, as in *Figure 3*. For visualization purposes, only electrodes that are more likely to be assigned to V1–V3 than any other map are plotted, even though the probabilistic assignment (e.g. for panel c) included additional electrodes. (**b**) pRFs were normalized by rotation (subtracting the broadband pRF polar angle from both the broadband and alpha pRF) and then scaling (dividing the eccentricity and size of both pRFs by the broadband pRF eccentricity). This puts the broadband pRF center at (0,1) for all electrodes, indicated by an 'x.' Within this normalized space, the average pRF across electrodes was computed 5,000 times (bootstrapping across electrodes). 100 of these averages are indicated as dotted lines, and the average across all 5000 bootstraps is indicated as a solid line. (**c**) Each panel shows the relation between broadband and alpha pRF parameters in 2-D histograms. 53 electrodes were randomly sampled with replacement 5000 times (bootstrapping), and each time they were selected, they were probabilistically assigned to a visual area. The colormap indicates the frequency of pRFs in that bin (see *Comparison of alpha and broadband pRFs* in Methods 4.7). Note that the phases in the horizontal and vertical axes are wrapped to show their circular relations. The middle and right panels show the relation of eccentricity and pRF sizes, respectively. Yellow ellipses indicate the 1-sd covariance ellipses. (**d**) The plot shows pRF size vs eccentricity, both for broadband (blue) and alpha (red). As in panel (**c**), the 53 electrodes were bootstrapped 5000 times. The regression lines are the average across bootstraps, and the shaded regions are the 16th to 84th percentile evaluated at each eccentricity. Electrodes that are more likely to be assigned to V1–V3 than any other maps are plotted as dots. (see *Figure 5—figure supplement 1* for electrodes in dorsolateral maps). See *makeFigure5_5* S1.m.

The online version of this article includes the following figure supplement(s) for figure 5:

**Figure supplement 1.** Relations of alpha and broadband population receptive fields (pRFs) in dorsolateral.

**Figure supplement 2.** Estimated population receptive field (pRF) size vs cross-validated variance explained.

of the alpha pRF might be systematically larger than the broadband eccentricity, even if the two signals originate from the same retinotopic map.

Our results also indicate that pRF size is much larger for alpha than for broadband. Because the comparison between alpha and broadband pRF size has such a steep slope (*Figure 5c*, right), one might wonder whether the size of the alpha pRFs vary in any meaningful way. Contrary to this possibility, we find that for both broadband pRFs and alpha pRFs, the pRF size systematically increases with eccentricity (*Figure 5d*, *Figure 5—figure supplement 1d*), consistent with known properties of visual neurons (*Newsome et al., 1986*) and with fMRI data (*Dumoulin and Wandell, 2008*; *Kay et al., 2013a*). The slopes of the size vs. eccentricity functions are about three times larger for alpha than for broadband (slopes: 0.53 vs 0.15 in V1–V3; 1.17 vs 0.40 in dorsolateral, for alpha vs broadband).

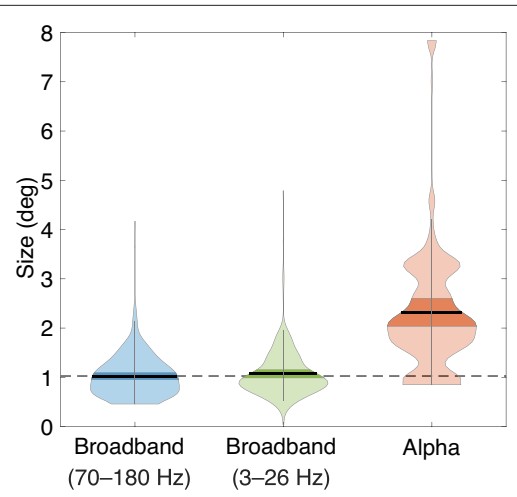

**Figure 6.** population receptive field (pRF) sizes in broadband and alpha. Violin plots of pRF sizes with 5000 bootstraps, estimated from high-frequency broadband, low-frequency broadband, and alpha. Black horizontal lines and dark color regions within violins indicate means ± 1 standard deviation of the mean across the 5000 samples. A dotted horizontal line indicates the mean of high-frequency broadband pRF sizes. For the full power spectra showing the broadband and alpha responses, see *Figure 6—figure supplement 1*. See *makeFigure6_6* S2.m.

The online version of this article includes the following figure supplement(s) for figure 6:

**Figure supplement 1.** Power in the broadband response and alpha oscillation.

**Figure supplement 2.** Low broadband model comparisons.

## The difference in pRF size is not due to a difference in temporal frequency

The larger pRF size for the alpha oscillation is not explained by the fact that the alpha band (8–13 Hz) is lower frequency than the band used to estimate the broadband signal (70–180 Hz). For V1 to V3, broadband power extended into low frequencies (*Figure 6—figure supplement 1*), enabling us to quantify the spatial tuning of the low-frequency portion of the broadband response (3–26 Hz). Unlike the alpha pRFs, which had negative gain and a large size, the pRFs for the low frequency broadband response had a positive gain and a small size, similar to the high-frequency broadband response (sizes: 1.0 vs 1.1 vs 2.3 deg, for high-frequency broadband vs low-frequency broadband vs alpha; *Figure 6*, *Figure 6—figure supplement 2*). The difference in size between low-frequency broadband and alpha pRFs suggests that the negative alpha modulations are not artifacts resulting from the baseline correction applied in our model-based approach.

## The accuracy of the alpha pRFs depends on the baseline correction

We used a model-based approach to separate the alpha oscillatory power from the broadband response because the two responses overlap in temporal frequency. This model adjusts the baseline for computing alpha oscillatory power. Had we not used this approach, and instead used a more traditional method of computing band-limited power without a baseline correction, we would have obtained different values for the alpha responses and inaccurate pRF fits. The difference between the two methods is especially clear in V1–V3, where the broadband response extends into the low frequencies (*Figure 6—figure supplement 1*). To clarify the advantage of the model-based approach, we compared the two types of pRF fits for all visually responsive electrodes in V1–V3 (based on maximum probability of map assignment: *Figure 7*). For these comparisons, we allowed the gain to be positive or negative. First, the cross-validated variance explained was about 4 times higher when correcting for the baseline (43% vs 10%, median). Second, as expected, the baseline-corrected approach consistently results in negative gain (29 out of 31 electrodes), meaning that alpha oscillations are suppressed by stimuli in the receptive field. In contrast, with no baseline correction, there is a mixture of positive and negative gain (14 negative, 17 positive), complicating the interpretation. Third, with baseline correction, pRF size increases with eccentricity, but without baseline correction, it does not. These analyses confirm the importance of disentangling the alpha oscillation from the broadband response.

## Alpha oscillations are coherent across a larger spatial extent than broadband signals

The large pRFs for alpha suppression might result from a limit to the spatial resolution at which alpha oscillations can be controlled. If, for example, alpha oscillations are up- or down-regulated with a point spread function of ~5 or 10 mm of cortex, then alpha pRFs would necessarily be large. We assessed

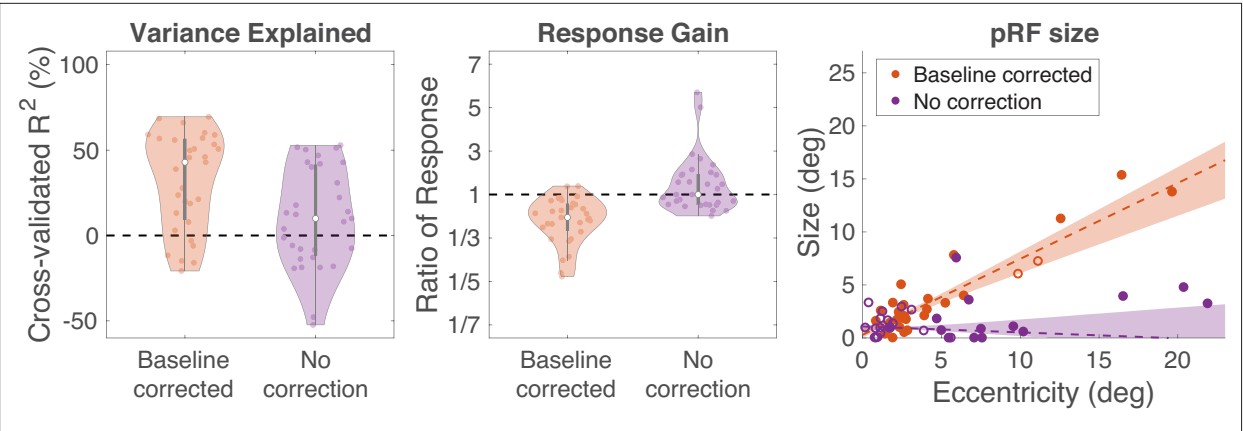

**Figure 7.** The advantage of baseline correction on alpha population receptive fields (pRFs). We compared pRF solutions for alpha using our spectral model ('baseline corrected') vs by computing power within the alpha band ('no correction'). **Left**: Variance explained. The white dots are the median and the gray bars are the interquartile range. Each colored dot is one electrode and the shaded regions are smoothed histograms. **Middle**: Response gain, plotted in the same manner as variance explained. Gain is quantified as the maximum or minimum value in the pRF fitted time series (maximal for electrodes with positive gain, minimal for electrodes with negative gain). A value above 1 means that the response increased relative to baseline (positive gain). A value below 1 means the response decreased (negative gain). **Right**: pRF size vs eccentricity. Electrodes are colored by the sign of the gain (filled for negative, open for positive). These tendencies are consistent across individual patients (*Figure 7—figure supplement 1*). The shadings represent 68% bootstrapped confidence intervals with 1000 resamples. See *makeFigure7_7* S1.m.

The online version of this article includes the following figure supplement(s) for figure 7:

**Figure supplement 1.** A comparison of alpha population receptive fields (pRFs) with and without baseline correction in individual patients.

the spatial resolution of the alpha oscillation by measuring coherence of the signal across space. Specifically, we measured coherence at the alpha frequency for electrode pairs in two high-density grids (Patients 8 and 9; See *Figure 8—figure supplements 1 and 2* for pRF parameters on the two high-density grids). These grids have 3 mm spacing between electrodes, about 10 times the density of standard grids (1 cm spacing). For comparison, we also measured the coherence between the same electrode pairs averaged across the frequencies we used for calculating broadband responses (70 to 180 Hz).

We find two clear patterns as shown in *Figure 8*. First, as expected, coherence declines with distance. The alpha coherence is about 0.6 between neighboring electrodes, declining to a baseline level of about 0.35 by about 1 cm. Second, the coherence was higher at the alpha frequency than in the broadband frequencies, especially between neighboring electrodes. The pattern is the same whether the coherence is computed only from trials in which the bar position overlapped pRF, or only from trials in which it did not overlap the pRF, or both (data not shown). Overall, the larger coherence at the alpha frequency is consistent with coarser spatial control of this signal (in mm of cortex), and larger pRFs (in deg of visual angle). The highest coherence is at the peak alpha frequency (*Figure 8— figure supplement 3*). This suggests that the elevated coherence is not simply an artifact of greater volume conduction at low frequencies.

## Discussion

Our main two findings are that visual cortex alpha suppression is specific to stimulus location, as measured by its pRF, and that alpha pRFs are more than twice the size of broadband pRFs. This supports that alpha suppression in and near the locations of the driven response increases cortical excitability. This has implications for visual encoding in cortex, perception, and attention, with several close parallels between the alpha pRF and exogenous (stimulus-driven) attention.

### What is the function of the large alpha pRFs?

The alpha oscillation was originally characterized as an 'idling' or default cortical state, reflecting the absence of sensory input or task demand (*Adrian and Matthews, 1934*; *Barry et al., 2007*; *Berger, 1929*; *Klimesch et al., 1998*; *Matousek and Petersén, 1983*). However, more recent work

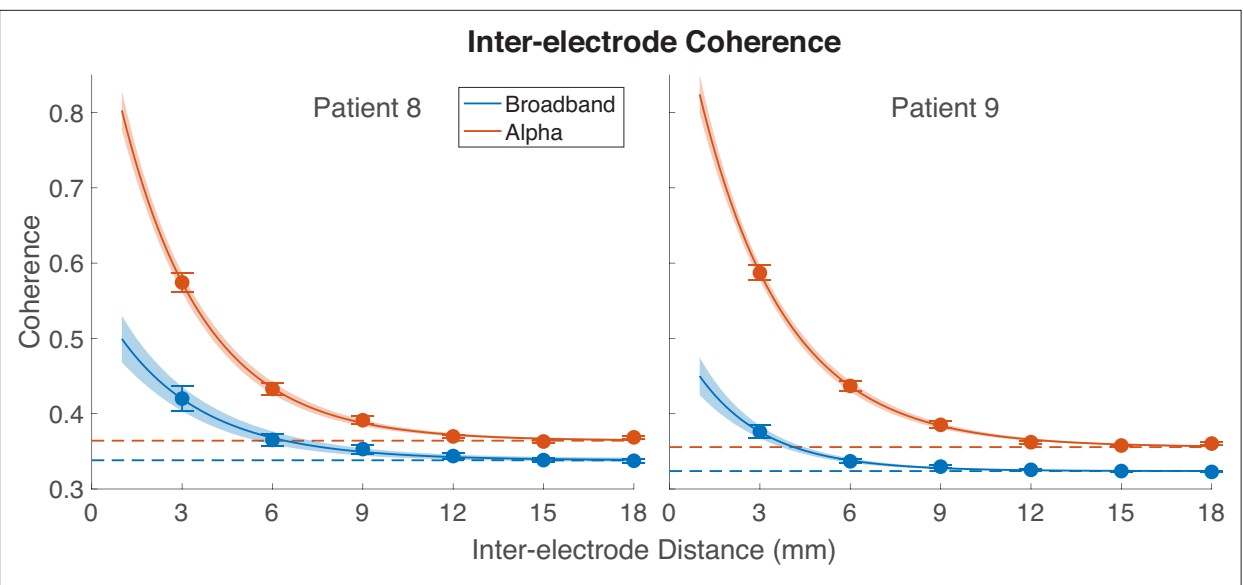

**Figure 8.** Inter-electrode coherence across distances in high-density grids. Coherence between electrode pairs was computed within each 1 s epoch and then averaged epochs. Seed electrodes were those which had a reliable population receptive field (pRF) fit, and these seed electrodes were compared to all other electrodes on the grid. The curved lines indicate the average fit of an exponential decay function across 5000 bootstraps; shading indicates ±1 standard deviation of bootstraps; dashed lines indicate the baselines the coherence converges to. See *Figure 8—figure supplement 3* for coherence as a function of frequency. See makeFigure8.m.

The online version of this article includes the following figure supplement(s) for figure 8:

**Figure supplement 1.** Population receptive field (pRF) parameters from high-density grid in Patient 8.

**Figure supplement 2.** population receptive field (pRF) parameters from high-density grid in Patient 9.

**Figure supplement 3.** Inter-electrode coherence in the high-density grids across 1–200 Hz.

supports the idea that alpha oscillations act as a gate, actively suppressing neural signals by reducing cortical excitability (*Bastos et al., 2020*; *Jensen and Mazaheri, 2010*; *Klimesch et al., 2007*; *Mazaheri and Jensen, 2010*; *Van Diepen et al., 2019*) via 'pulsed inhibition' (hyperpolarization followed by rebound). This pulsed inhibition explains why cortical excitability depends on both alpha oscillation *magnitude* (*Kelly et al., 2006*; *Rihs et al., 2007*; *Worden et al., 2000*) and *phase* (*Osipova et al., 2008*; *Spaak et al., 2014*): when the oscillations are large, there is one phase when cortex is most excitable and one phase when it is least excitable (phase dependence), and when the oscillations are small, cortex is on average more excitable (magnitude dependence). When coupled to the large pRFs, this predicts that the onset of a focal visual stimulus will cause an increase in cortical excitability, spreading beyond the region that responds directly, i.e., with broadband field potentials and spiking (*Figure 9*). A transient increase in excitability spreading beyond the stimulus is conceptually similar to stimulus-driven ('exogenous') spatial attention, in which a visual cue leads to faster response times, increased discriminability, higher perceived contrast (*Carrasco, 2011*; *Downing and Pinker, 1985*; *Shulman et al., 1986*), and increased neural response, as measured by single unit spike rates (*Wang et al., 2015a*) and BOLD fMRI (*Dugué et al., 2020*).

The relationship may be more than conceptual, however, as we find that the spatial profiles of attention, measured behaviorally (*Downing and Pinker, 1985*; *Shulman et al., 1986*), and of the alpha pRF are quite similar (*Figure 10*), with three close parallels. For both measures, the spatial spread (1) increases with eccentricity, (2) is asymmetric, spreading further into the periphery than toward the fovea, and (3) shows center/surround organization, with the attentional effect changing from a benefit to a cost and the alpha response changing from suppression to enhancement. The timing also matches, since exogenous attention has a maximal effect at about 100ms (*Nakayama and Mackeben, 1989*; *Wang et al., 2015a*), the duration of an alpha cycle.

While many studies have proposed that attention, whether stimulus-driven (exogenous) or internal (endogenous), influences the alpha oscillation (*Foster et al., 2017*; *Kelly et al., 2006*; *Popov et al., 2019*; *Rihs et al., 2007*; *Samaha et al., 2016*; *Sauseng et al., 2005*; *Worden et al., 2000*), our

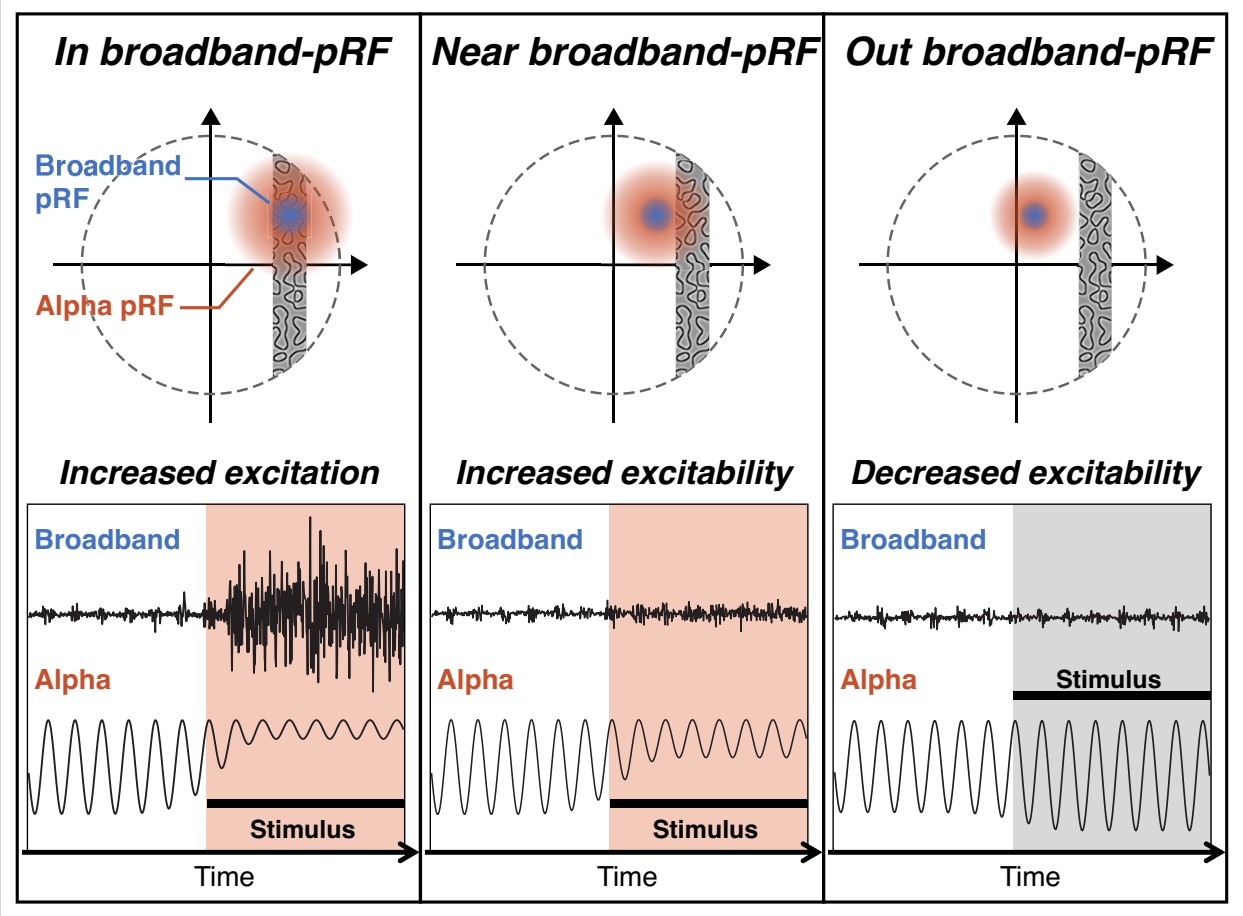

**Figure 9.** Schematics of neural excitation and excitability changes induced by visual stimulation. The upper panels show stimulus position relative to the broadband and alpha population receptive field (pRFs) for a given electrode. The lower panel shows predicted alpha and broadband responses from that electrode, with stimulus onset indicated by the black horizontal bar. In the left and middle panels, the stimulus causes the alpha oscillations to be suppressed and cortex to become more excitable (red shading). In the right panel, the alpha oscillation increases slightly and cortex becomes less excitable (gray shading).

proposal flips the causality. We propose that the stimulus causes a change in the alpha oscillation and this in turn causes increased neural responsivity and behavioral sensitivity, the hallmarks of covert spatial attention. The spatial spread of the alpha oscillation can be traced in part to the neural mechanisms that generate it in the thalamus (see section 3.2). Top-down attention may capitalize on some of the same mechanisms, as feedback to the LGN can modulate the alpha-initiating neural populations, then resulting in the same effects of cortical excitability in visual cortex that occur from bottom-up spatial processing. This is consistent with findings that endogenous attention is accompanied by local reductions in alpha oscillations (*Popov et al., 2019*; *Sauseng et al., 2005*; *Worden et al., 2000*).

This link to attention contrasts with the proposal that the alpha oscillation in visual cortex reflects surround suppression (*Harvey et al., 2013*). We think the alpha oscillation is not tightly linked to surround suppression. First, the alpha pRF is negative, and alpha tends to be inhibitory (*Bastos et al., 2020*; *Jensen and Mazaheri, 2010*; *Klimesch et al., 2007*; *Mazaheri and Jensen, 2010*; *Van Diepen et al., 2019*), meaning that visual stimuli within the alpha pRF disinhibit cortex, the opposite of surround suppression. An excitatory surround (alpha suppression) and a suppressive surround can coexist if they differ in timing and feature tuning. Surround suppression is relatively fast, disappearing 60ms after surround stimulus offset (*Bair et al., 2003*), whereas the ~10 Hz alpha oscillation is likely to be modulated at a slower time scale. Furthermore, surround suppression is feature-specific (*Bair et al., 2003*), whereas the alpha oscillation, being coherent over several mm of cortex, is unlikely to be limited only to cells whose tuning match the stimulus. Nonetheless, surround suppression may

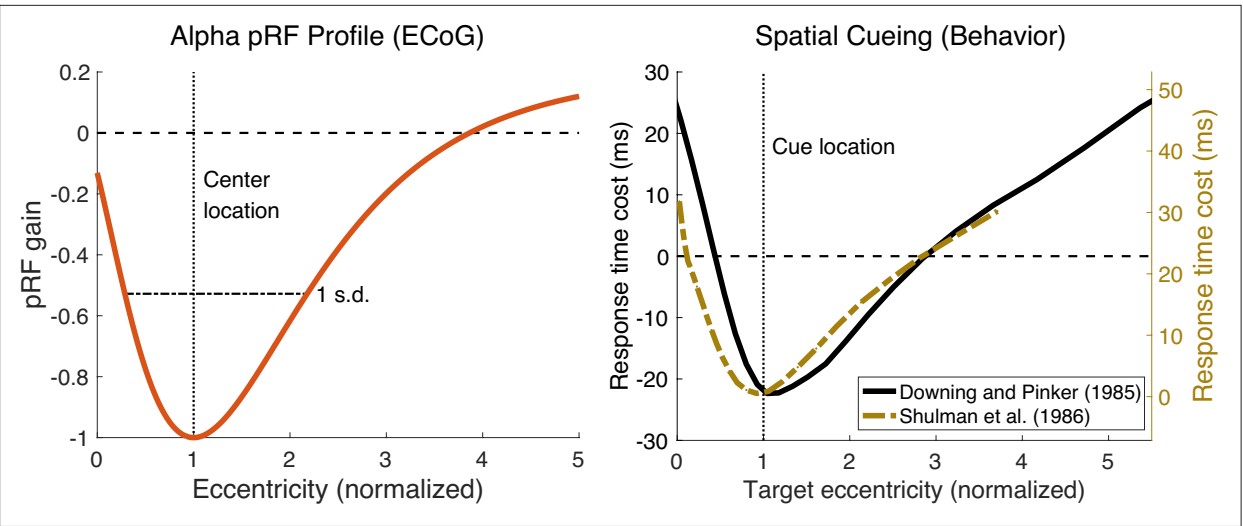

**Figure 10.** Asymmetric profile of alpha population receptive field (pRF) and asymmetric effect of spatial attention. The left panel shows an asymmetric profile of alpha pRF across normalized eccentricity in V1–V3. The right panel shows asymmetric effects of attention on normalized cue–target locations. Black line represents response time changes induced by endogenous attention from ***Downing and Pinker, 1985***,, relative to the response time without any attentional cue. The gold line represents response time changes induced by exogenous attention from ***Shulman et al., 1986***,, relative to the response time when targets appeared at the attended locations. Results from the Shulman et al. paper with foveal cue locations at 0.5 degree eccentricity were excluded from the plot because after normalizing, the data from this condition has no overlap with the other conditions perhaps because at such a small eccentricity, a minor error in attended location makes a large difference on a normalized scale. The experimental results for each cue location are shown in ***Figure 10—figure supplement 1***. See makeFigure10.m.

The online version of this article includes the following figure supplement(s) for figure 10:

**Figure supplement 1.** Response time costs in previous spatial attention experiments.

be linked to neural oscillations, albeit at a higher frequency range (***Hermes et al., 2019***; ***Ray et al., 2013***).

Other studies have proposed that retinotopically specific alpha oscillations are related to oculo-motor behavior (***Popov et al., 2021a***; ***Popov et al., 2021b***). Supporting evidence includes the obser-vations that oscillations decrease with saccade initiation (***Popov et al., 2021b***) and that saccades can be phase-locked to the alpha rhythm (***Drewes and VanRullen, 2011***; ***Staudigl et al., 2017***). In our study, participants maintained fixation at the center of the screen during receptive field mapping, dissociating modulations in the alpha oscillation from large eye movements. Nonetheless, there are many parallels, and likely shared circuitry, between spatial attention and eye movements (***Moore and Fallah, 2001***; ***Shepherd et al., 1986***).

An alternative interpretation of our results is that the larger alpha pRFs are an artifact of lower signal-to-noise ratio. We think this is unlikely for three reasons. First, simulations of pRFs in which the ground truth is known show no systematic increase in pRF size with more noise (***Lerma-Usabiaga et al., 2020***, their Figure 8). Second, adding substantial noise to real fMRI data had little to no effect on pRF size (***Le et al., 2017***, their Figure 6). Third, the larger size of the alpha pRFs is preserved when regressing size vs variance explained (***Figure 5—figure supplement 2***).

We focused on the spatial specificity of alpha oscillations on the cortical surface. A separate line of research has also examined the specificity of alpha oscillations in terms of cortical depth and inter-areal communication. One claim is that oscillations in the alpha and beta range play a role in feedback between visual areas, with evidence from nonhuman primates for specificity of alpha and beta oscil-lations in cortical depth (***Bastos et al., 2020***; ***Bollimunta et al., 2011***). To our knowledge, no one has examined the spatial specificity of the alpha oscillation across the cortical surface and across cortical depth in the same study, and it remains an open question as to whether the same neural circuits are involved in both types of effects.

## The importance of separating oscillatory and non-oscillatory signals

Separating the alpha power into oscillatory and broadband components was motivated by the different biological origins of the two signals. Recent work has identified a population of high threshold

thalamocortical neurons that generates bursts in the alpha or theta frequency range (depending on membrane potential) (*Hughes and Crunelli, 2007*; *Hughes et al., 2011*; *Lorincz et al., 2009*). These bursts are synchronized by gap junctions. These neurons themselves appear to be the pacemakers, generating the alpha rhythm in the LGN and transmitting it to V1. This explains the rhythmic nature of the signal (bursting), the coherence across space (synchrony by gap junctions), and the occipital locus of alpha (thalamocortical generators). The fact that this cell population is located in the LGN does not imply that the alpha oscillation is only based on feedforward signals. Feedback signals from V1 to LGN via corticothalamic cells can modulate the frequency and amplitude of the alpha generating thalamo-cortical cells (*Hughes and Crunelli, 2007*). There may also be intracortical circuits that modulate the alpha oscillation (*Bollimunta et al., 2011*; *Clayton et al., 2018*).

The broadband response differs in several ways. First and most obvious, the signal is not limited to a narrow frequency range (*Hermes et al., 2015*; *Hermes et al., 2017*; *Hermes et al., 2019*; *Miller et al., 2009a*; *Winawer et al., 2013*), and hence is not generally considered an oscillation (*Miller et al., 2014*). Therefore, functions which are hypothesized to depend specifically on oscillations should likely not be imputed to this signal, such as a behavioral bias during certain phases of the oscillation (*Busch et al., 2009*; *Mathewson et al., 2009*; *Spaak et al., 2014*) or long range synchronization of signals (*Palva and Palva, 2011*; *Palva and Palva, 2012*). Second, broadband signals are found throughout the brain (*Kupers et al., 2018*; *Manning et al., 2009*; *Miller et al., 2014*), likely arising from non-oscillatory neural generators (*Bédard et al., 2006*; *Miller et al., 2009a*), whereas the alpha rhythm is largest in the occipital lobe. Finally, the broadband signal measured at the electrode has lower amplitude because the generators are not synchronized across space.

The motivation to separate the signals was confirmed by the results. Compared to the frequency band method (no decomposition), the decomposition method resulted in higher variance explained, more consistent direction (negative pRFs rather than a mixture of positive and negative), and a tighter relationship between pRF eccentricity and size. The frequency band method has poorer results because it conflates two signals with different properties.

The difference between the methods likely explains some discrepancies across studies. *Klink et al., 2021* fit pRF models to electrode data from macaque primary visual cortex in many frequency bands, including alpha, without removing the broadband signal. They found no systematic relationship between pRF size and eccentricity and found that the gain was positive for some electrodes and negative for others. The interpretation was that some locations in the visual field show suppression and others excitation. Our results suggest that it is more likely that most or all locations show both responses, alpha suppression and broadband increase, with either of the two signals sometimes being larger. Second, *Takaura et al., 2016* estimated receptive fields in macaque in several frequency bands, including alpha. They found that alpha power tended to increase by visual stimulation, in contrast to our results. We speculate that visual stimuli decreased alpha oscillations, but that this decrease was masked by a larger broadband power increase. Finally, *Harvey et al., 2013* examined spatial selectivity of human ECoG responses in the alpha band. They fit pRF models to the broadband response and then measured alpha power for stimulus positions relative to the broadband pRFs. They reported that alpha power increased in V1 electrodes when stimuli were outside the pRF, and that there was no change in alpha power when the stimulus overlapped the broadband pRF. The combination of these two results led to an interpretation that the alpha response is specifically related to surround suppression. Our results confirm their first observation that alpha power increases when the stimulus is outside the pRF (as reflected in our Difference of Gaussians model); however, in contrast to their report, we find strong alpha suppression when the stimulus is in the pRF center. The discrepancy is reconciled by the approach. We, too, find that total alpha power is approximately unchanged by the stimulus in the pRF center, but the model-based approach shows that this is due to cancellation from an increased broadband response and a reduced alpha oscillation, rather than to the alpha oscillation not changing.

A similar decomposition approach was used in prior ECoG studies to separate broadband power from steady state visual evoked potentials (*Winawer et al., 2013*) and from narrowband gamma oscillations (*Hermes et al., 2015*; *Hermes et al., 2017*; *Hermes et al., 2019*). A generalization of this approach has also been implemented in a toolbox for separating broadband and narrowband signals (*Donoghue et al., 2020*). These approaches are premised on the idea that different signals arise from different neurobiological causes and may be modulated independently by stimulus or task. This

contrasts with the tradition of separating the field potential into frequency bands, where band-limited power is often computed without identifying a spectral peak and without removing non-oscillatory contributions.

One potential concern about our approach might be circularity: is it possible that we found the alpha and broadband signals to have opposite signs and similar pRF locations because there is in fact, only one independent neural response, which we projected onto two measures? We think this is not the case. First, the broadband and the alpha power changes fitted by the pRF model were computed in different frequency ranges so that they were mathematically independent: the broadband was computed in high frequency range (70–180 Hz) and the alpha was decomposed from the low-frequency broadband response in 3–26 Hz (or 3–32 Hz). Second, if the two responses arose from one independent signal, then both the center locations and the sizes of alpha and broadband pRFs would be the same, but the alpha pRFs were much larger than the broadband pRFs.

The advantage of our approach is not limited to the visual system. Alpha suppression is observed in other cortical areas (*Clayton et al., 2018*), as are broadband responses (*Miller et al., 2014*), and hence the need for spectral modeling is likely to be widespread.

## How broad is broadband?

Since Crone et al.'s (1998) observations of task-related power increases at high frequency (70–100 Hz, above the typical gamma band), there has been a great deal of interest in these high frequency signals. Unlike the lower frequency gamma oscillations studied previously (*Eckhorn et al., 1988*; *Gray and Singer, 1989*), the signals above 70 Hz had no clear peak. Crone et al. speculated that this was due to the ECoG signal pooling across multiple oscillatory circuits with different peak frequencies. Signal changes above 70 Hz have been referred as 'high gamma' (*Bartoli et al., 2019*; *Brovelli et al., 2005*; *Canolty et al., 2006*; *Ray and Maunsell, 2011*), implicitly conveying that the signals are similar to gamma oscillations, but just at a higher frequency.

In contrast, *Miller et al., 2009a* proposed that these signals, although appearing to be concentrated in high frequencies, do not in fact arise from neural oscillations at all. They interpreted these signals as reflecting broadband neural activity with no specific time scale, spanning the whole spectrum but obscured at lower frequencies by changes in narrowband phenomena (*Miller et al., 2009b*). Such task-related broadband power increases have been found in human intracranial electrodes over V1 to V3 (*Winawer et al., 2013*) and in human microelectrode recordings of local field potentials (*Manning et al., 2009*). Our results confirm their findings as the broadband increase is observed from as low as 3 Hz in V1–V3. The broadband responses in low-frequency (3–26 Hz) and high-frequency band (70–180 Hz) showed similar response patterns to the pRF stimuli, as reflected in similar pRF model parameters, suggesting they have a common cause.

There may, however, be variability across cortical locations (and across people) in the neural circuit properties that generate broadband responses. We find a difference in the pattern between V1–V3 and the dorsolateral maps. In V1–V3, the stimulus-related broadband elevation is clear all the way down to 3 Hz or lower, whereas in dorsolateral cortex it is clear only down to about 25 Hz. The difference could reflect fundamental differences in the type of field potentials that can be generated from the two regions, or could depend on how well matched the stimuli are to the tuning of the brain areas: the simple contrast patterns used for pRF mapping more effectively drive the early visual field maps than higher level areas (e.g. producing twice the percent signal change in high broadband signals). The human microelectrode recordings (*Manning et al., 2009*) indicate that broadband power extending as low as 2 Hz can be measured across many brain regions, including frontal, parietal, and occipital cortices as well as hippocampus and amygdala. A further consideration is that broadband power changes might entail changes in the exponent of the spectral power function in addition to a change in the scaling (*Donoghue et al., 2020*). In either case, however, it is likely that the power changes in high frequency regions reported here and elsewhere (e.g. *Miller et al., 2014*) reflect processes which are not oscillatory and hence not limited to a narrow range of frequencies. And because the model-based method of quantifying alpha oscillations shows a clear advantage for some regions (V1–V3) and no disadvantage for others, it is a more interpretable and general method than simply measuring the power at a particular frequency.

## Conclusion

With high spatial precision measurements, we have demonstrated that there are two independent responses to visual stimulation in the alpha band: suppression of alpha oscillations and elevation of

broadband power. Separating the two responses was essential for accurately fitting pRF models to the alpha oscillation. The large, negative alpha pRF indicates that visual stimulation suppresses alpha oscillations and likely increases cortical excitability over a large cortical extent (larger than the region with broadband and spiking increases). Together, these results predict many of the effects observed in spatial cueing experiments, thus providing further links between the alpha oscillation and spatial attention. More generally, alpha oscillation is likely to play a role in how cortex encodes sensory information.

## Methods

The dataset was collected as part of a larger project funded by the NIH (R01MH111417). The methods for collecting and preprocessing the visual ECoG data have been described recently (*Groen et al., 2022*). For convenience, data collection methods below duplicate some of the text from the methods section of *Groen et al., 2022* with occasional modifications to reflect slight differences in participants, electrode selection, and pre-processing.

### Data and code availability

All data and code are freely available. We list the main sources of data and code here for convenience: The data are shared in BIDS format via Open Neuro (https://openneuro.org/datasets/ds004194). The analysis depends on two repositories with MATLAB code: one for conversion to BIDS and pre-processing (ECoG_utils, https://github.com/WinawerLab/ECoG_utils), and one for analysis of the pre-processed data (ECoG_alphaPRF, https://github.com/KenYMB/ECoG_alphaPRF). The analysis also has several dependencies on existing public tools, including analyzePRF (*Kay et al., 2013a*; https://github.com/cvnlab/analyzePRF; *Kay, 2025*), FieldTrip (*Oostenveld et al., 2011*; https://github.com/fieldtrip/fieldtrip; *Oostenveld and Schoffelen, 2025*), and FreeSurfer (http://freesurfer.net). Within the main analysis toolbox for this paper (ECoG_alphaPRF), there is a MATLAB script to re-generate each of the main data figures in this paper, called '*makeFigure1.m*,' '*makeFigure2.m*,' etc. Sharing both the raw data and the full set of software tools for analysis and visualization facilitates computational reproducibility and enables inspection of computational methods to a level of detail that is not possible from a standard written Methods section alone.

All code used for generating the stimuli and for experimental stimulus presentation can be found at https://github.com/BAIRR01/BAIR_stimuli (*Winawer and Montenegro, 2025*) and https://github.com/BAIRR01/vistadisp (*BAIRR01, 2022*).

### Participants

ECoG data were recorded from 9 participants implanted with subdural electrodes for clinical purposes. Data from seven patients were collected at New York University Grossman School of Medicine (NYU), and from two patients at the University Medical Center Utrecht (UMCU). The participants gave informed consent to participate, and the study was approved by the NYU Grossman School of Medicine Institutional Review Board and the ethical committee of the UMCU. Detailed information about each participant and their implantation is provided in *Table 1*. Two patients (p03 and p04) were not included because they did not participate in the pRF experiment which we analyzed for the purpose of the current study.

### ECoG recordings
#### NYU
Stimuli were shown on a 15 in. MacBook Pro laptop. The laptop was placed 50 cm from the participant's eyes at chest level. Screen resolution was 1280×800 pixels (33×21 cm). Prior to the start of the experiment, the screen luminance was linearized using a lookup table based on spectrophotometer measurements (Cambridge Research Systems). ECoG data were recorded using four kinds of electrodes: Standard size grids, linear strips, depth electrodes, and high-density grids, with the following details.

- Standard grids (8×8 arrays) and strips (4–12 contact linear strips) were implanted, consisting of subdural platinum-iridium electrodes embedded in flexible silicone sheets. The electrodes

**Table 1.** Overview of patient data included in this dataset.

The patient number and subject code for BIDS do not always match because the nine subjects with population receptive field (pRF) data are a subset of 11 subjects from a larger BIDS dataset (https://openneuro.org/datasets/ds004194). Two of the 11 patients did not take part in pRF experiments.

| Patient | BIDS Code | Age | Sex | Site | PRF runs | Hemi | Implantation | Coverage | # of Visual | Matching areas with maximum probability |
|---|---|---|---|---|---|---|---|---|---|---|
| Patient 1 | p01 | 30 | F | UMCU | 4 | L | strip, depth | lat-occ, lat-par, vent-temp | 8 | PHC, V3AB, LO, TO |
| Patient 2 | p02 | 18 | F | UMCU | 2 | R | grid, strip | med-occ, lat-occ, lat-temp, vent-temp | 20 | V1, V2, V3, hV4, V3AB, TO, IPS |
| Patient 3 | p05 | 31 | F | NYU | 4 | R | 2×grid, strip, depth | lat-occ, lat-par, frontal, ant-temp | 21 | V2, V3, hV4, LO, TO, IPS |
| Patient 4 | p06 | 19 | F | NYU | 2 | L | grid, strip, depth | med-occ, lat-occ, lat-par, ant-temp, frontal | 28 | V1, V2, V3, hV4, V3AB, LO, TO, IPS |
| Patient 5 | p07 | 41 | F | NYU | 4 | L | grid, strip, depth | lat-occ, lat-par, med-occ, ant-temp, frontal | 20 | V1, V2, V3, V3AB, TO, IPS |
| Patient 6 | p08 | 47 | F | NYU | 2 | R | grid, strip, depth | lat-par, ant-temp | 6 | LO, TO, IPS |
| Patient 7 | p09 | 25 | M | NYU | 4 | L,R | bilateral grid, strip, depth | lat-occ, lat-par | 8 | LO, TO, IPS |
| Patient 8 | p10 | 23 | M | NYU | 6 | R | grid, HD grid, strip, depth | lat-occ, lat-par, post-vent | 130 | V2, V3, V3AB, LO, TO, IPS, SPL |
| Patient 9 | p11 | 19 | F | NYU | 6 | L | grid, HD grid, strip, depth | lat-occ, lat-par, post-vent | 93 | hV4, VO, V3AB, LO, TO, IPS |

had 2.3 mm diameter exposed surface and 10 mm center-to-center spacing (Ad-Tech Medical Instrument, Racine, Wisconsin, USA).
- Penetrating depth electrodes (1×8 or 1×12 contacts) were implanted, consisting of 1.1 mm diameter electrodes, 5- to 10 mm center-to-center spacing (Ad-Tech Medical Instrument).
- In two patients (Patient 8 and 9), there were small, high-density grids (16×8 array) implanted over lateral posterior occipital cortex. These high-density grids were comprised of electrodes with 1 mm diameter exposed surface and 3 mm center-to-center spacing (PMT Corporation, Chanhassen, Minnesota, USA).

Recordings were made using one of two amplifier types: NicoletOne amplifier (Natus Neurologics, Middleton, WI), bandpass filtered from 0.16 to 250 Hz and digitized at 512 Hz, and Neuroworks Quantum Amplifier (Natus Biomedical, Appleton, WI) recorded at 2048 Hz, bandpass filtered at 0.01–682.67 Hz and then downsampled to 512 Hz. Stimulus onsets were recorded along with the ECoG data using an audio cable connecting the laptop and the ECoG amplifier. Behavioral responses were recorded using a Macintosh wired external numeric keypad that was placed in a comfortable position for the participant (usually their lap) and connected to the laptop through a USB port. Participants self-initiated the start of the next experiment by pushing a designated response button on the number pad.

## UMCU

Stimuli were shown on an NEC MultiSync E221N LCD monitor positioned 75 cm from the participant's eyes. Screen resolution was 1920×1080 pixels (48×27 cm). Stimulus onsets were recorded along with the ECoG data using a serial port that connected the laptop to the ECoG amplifier. As no spectro-photometer was available at the UMCU, screen luminance was linearized using the built-in gamma table of the display device. Data were recorded using the same kinds of electrodes as NYU: Standard grids and strips, and depth electrodes, and using a MicroMed amplifier (MicroMed, Treviso, Italy) at 2048 Hz with a low-pass filter of 0.15 Hz and a high-pass filter of 500 Hz. Responses were recorded with a custom-made response pad.

## pRF stimulus

Visual stimuli were generated in MATLAB 2018b. Stimuli were shown at 8.3 degrees of visual angle (16.6 degrees stimulus diameter) using Psychtoolbox-3 (http://psychtoolbox.org/) and were presented at a frame rate of 60 Hz. Custom code was developed in order to equalize visual stimulation across the two recording sites as much as possible; for example, all stimuli were constructed at high resolution (2000×2000 pixels) and subsequently downsampled in a site-specific manner such that the stimulus was displayed at the same visual angle at both recording sites (see *stimMakePRFExperiment.m* and *bairExperimentSpecs.m* in BAIRR01/BAIR_stimuli). Stimuli consisted of grayscale band-pass noise patterns that were created following procedures outlined in *Kay et al., 2013a*. Briefly, the pattern stimuli were created by low-pass filtering white noise, thresholding the result, performing edge detection, inverting the image polarity such that the edges are black, and applying a band-pass filter centered at three cycles per degree (see *createPatternStimulus.m*).

All stimuli were presented within bar apertures; the remainder of the display was filled with neutral gray. This stimulus has been shown to effectively elicit responses in most retinotopic areas (*Kay et al., 2013b*). The width of the bar aperture was 2 degrees of visual angle, which was 12.5% of the full stimulus extent. The bar aperture swept across the visual field in eight directions consisting of 28 discrete steps per sweep, 850 ms step duration. During each step, the bar stimulus was displayed for 500 ms followed by a 350 ms blank period showing a gray mean luminance image. The eight sweeps included two horizontal (left to right or right to left), two vertical (up to down or down to up) and four diagonal sweeps (starting from one of the four directions). For the diagonal sweeps, the bar was replaced with a blank for the last 16 of the 28 steps, so that the stimuli swept diagonally across the visual field from one side to the center in twelve steps followed by 13.6 s of blanks (16×850 ms). Each experiment included 224,850 ms steps plus 3 s of blank at the beginning and end, for a total of 196.4 s.

## Experimental procedure

All participants completed pRF mapping experiments in which they viewed visual stimuli for the purpose of estimating the spatial selectivity for individual electrodes (population receptive field

mapping *Dumoulin and Wandell, 2008*). In these experiments, participants were instructed to fixate on a cross located in the center of the screen and press a button every time the cross changed color (from green to red or red to green). Fixation cross color changes were created independently from the stimulus sequence and occurred at randomly chosen intervals ranging between 1 and 5 s. Participants completed one to three recording sessions. Each session included two pRF mapping experiments. The experimenter stayed in the room throughout the testing session. Participants were encouraged to take short breaks between experiments.

## ECoG data analysis

### Preprocessing

Data were preprocessed using MATLAB 2020b with custom scripts available at https://github.com/WinawerLab/ECoG_utils (see *Data and code availability* for access to data and computational methods to reproduce all analyses). Raw data traces obtained in each recording session were visually inspected for spiking, drift, or other artifacts. Electrodes that showed large artifacts or showed epileptic activity were marked as bad and excluded from analysis. Data were then separated into individual experiments and formatted to conform to the iEEG-BIDS format (*Holdgraf et al., 2019*). Data for each experiment were re-referenced to the common average across electrodes for that experiment, whereby a separate common average was calculated per electrode group (i.e. separate common average for each of the four electrode types-standard grid, high-density grid, strip, and depth electrodes, see *bidsEcogRereference.m*). The re-referenced voltage time series for each experiment were written to the BIDS derivatives directories.

### Electrode localization

Intracranial electrode arrays from NYU patients were localized from the post-implantation structural T1-weighted MRI images and co-registered to the preoperative T1-weighted MRI (*Yang et al., 2012*). Electrodes from UMCU patients were localized from the postoperative CT scan and co-registered to the preoperative T1-weighted MRI (*Hermes et al., 2010*). Electrode coordinates were computed in native T1 space and visualized on the pial surface reconstruction of the preoperative T1-weighted scan generated using FreeSurfer.

Cortical visual field maps were generated for each individual participant based on the T1-weighted scan by aligning the surface topology with a probabilistically defined retinotopic atlas derived from a retinotopic fMRI mapping dataset as shown in *Figure 1—figure supplement 1*; *Wang et al., 2015b*. We use the full probability map, rather than the maximum probability map, in order to account for the uncertainty in the mapping between electrodes location and visual field map in the patient's native T1 space. For each cortical vertex, the atlas contains the probability of being assigned to each of 25 maps. Electrodes were matched to the probabilistic atlases using the following procedure (see *bidsEcogMatchElectrodesToAtlas.m* in ECoG_utils): For each electrode, the distance to all the nodes in the FreeSurfer pial surface mesh was calculated, and the node with the smallest distance was determined to be the matching node. The atlas value for the matching node was then used to assign the electrode to a probability of belonging to each of the following visual areas: V1v, V1d, V2v, V2d, V3v, V3d, hV4, VO1, VO2, PHC1, PHC2, TO2, TO1, LO2, LO1, V3b, V3a, IPS0, IPS1, IPS2, IPS3, IPS4, IPS5, SPL1, or FEF. We then merged these 25 visual areas into 12 groups: V1 (V1v+V1 d), V2 (V2v+V2 d), V3 (V3v+V3 d), hV4, VO (VO1 +VO2), PHC (PHC1 +PHC2), TO (TO1 +TO2), LO (LO1 +LO2), V3AB (V3a+V3 b), IPS (IPS0–5), SPL (SPL1), or FEF.

The importance of assigning electrodes to areas probabilistically is that the uncertainty of assignment is reflected in the uncertainty of the estimates. Were we to uniquely assign each electrode to one area using maximum probability, we would likely overstate any conclusions about differences between maps.

To visualize the electrodes on the cortical surface we use a modified version of the Wang et al. full probability atlas. First, we select vertices that have a probability greater than 95% of not belonging to any of the retinotopic regions, and labeled as 'none.' Next, for the remaining vertices, we assigned the region with the highest probability, even if the probability for not belonging to one of the retinotopic regions was higher (e.g. *Figures 1 and 2*, *Figure 1—figure supplement 1*, *Figure 8—figure supplements 1 and 2*).

## Data epoching

The preprocessed data were analyzed using custom MATLAB code (https://github.com/KenYMB/ECoG_alphaPRF). First, a dataset was created by reading in the voltage time courses of each experiment for each participant from the corresponding BIDS derivatives folders. To combine the UMCU and NYU participants into a single dataset, the UMCU data were downsampled at 512 Hz to match the sample rate of the NYU data. Visual inspection of the data indicated an obvious delay in response onset for the UMCU participants relative to the NYU participants. The cause of the delay could not be tracked down but it was clearly artifactual. To correct the delay, UMCU data were aligned to the NYU data based on a cross-correlation on the average event-related potentials (ERPs) across all stimulus conditions from the V1 and V2 electrodes from three participants (1 UMCU, 2 NYU). The delay in stimulus presentation was estimated to be 72 ms (95% CI 63–85 ms by bootstrapping across a total of 18 electrode pairs), and stimulus onsets of the UMCU participants were shifted accordingly (*Groen et al., 2022*; see also the script s_determineOnsetShift UMCUvsNYU.m in https://github.com/irisgroen/temporalECoG; *Groen, 2023*). One NYU patient (Patient 3) had inaccurate trigger onset signals for each stimulus event due to a recording malfunction. The onset timings for this patient were reconstructed by using the first onset in each experiment (which were recorded correctly), and then assumed to begin every 850 ms as controlled by the stimulus computer. Any small error in these latency corrections should have no effect on our conclusions, as the evoked potentials were computed separately for each participant, and latency of evoked response played no role in our analysis.

The voltage data were then epoched for all trials between [–0.2 and 0.8] s relative to stimulus onset (see *ecog_prf_getData.m*). Baseline-correction was applied by subtracting the average pre-stimulus amplitude computed in [–0.2 and 0] s within each trial.

## Electrode selection and exclusion of noisy data

In two participants (10 and 11), there were high-density grids over lateral posterior cortex. All electrodes from these two grids were included in the pRF analyses (described below). In addition, we also included all standard grid or strip electrodes which were assigned to one of the visual areas with a minimum of 5% according to the Wang et al full probability atlas. Together, this resulted in 239 high-density electrodes and 131 standard grid or strip electrodes. After excluding four electrodes due to high noise (see paragraph below), the data selection resulted in a total of 366 electrodes from 9 participants. Of these 366 electrodes, 334 covered visual areas V1, V2, V3, hV4, VO, PHC, TO, LO, V3AB, IPS, and SPL. The other 32 electrodes belonged to one of the high-density grids but could not be assigned to a particular visual area.

Furthermore, we excluded epochs containing voltages that were large relative to the evoked signal. Specifically, for each electrode, we derived a distribution of the expected peak of the evoked signal by (1) converting voltage to power (squared voltage), (2) taking the maximum power from each epoch, excluding blanks, during the stimulus period (0–500 ms), and (3) fitting an inverse Gaussian distribution to these numbers. This yields one distribution for each electrode. Next, we computed the evoked potential by averaging the voltage time series across all epochs excluding blanks, and projected this out from every epoch. Finally, from the residual time series in each epoch, we compared the maximum power to the inverse Gaussian distribution for that electrode. If that value fell in the far tail, defined as the upper 0.01% of the area under the curve, then we excluded that epoch. This resulted in the exclusion of 0.8% of epochs (3422 out of ~430,000). Twelve electrodes showed unusually high variance across trials (over three standard deviations across electrodes) and were, therefore, excluded from further analysis. Furthermore, two runs from Patient 3 were not included in final analyses, as the patient broke fixation during these runs (see *ecog_prf_selectData.m*).

## Computing power spectral density

We computed power spectral density (PSD) for each electrode in each epoch. We assumed in our analysis that the alpha oscillation is bandpass, centered at about 8–13 Hz, and that the biophysical processes giving rise to the oscillation are distinct from those causing the low-pass visually evoked potential. The two signals overlap in frequency, however. Hence, in order to remove the influence of the evoked signal on estimates of alpha power (and other spectral responses) (*Hermes et al., 2015*; *Hermes et al., 2017*), we computed the averaged ERPs separately for epochs with stimulus presentation and for epochs without stimuli (blanks). We then regressed the ERP time series out from each

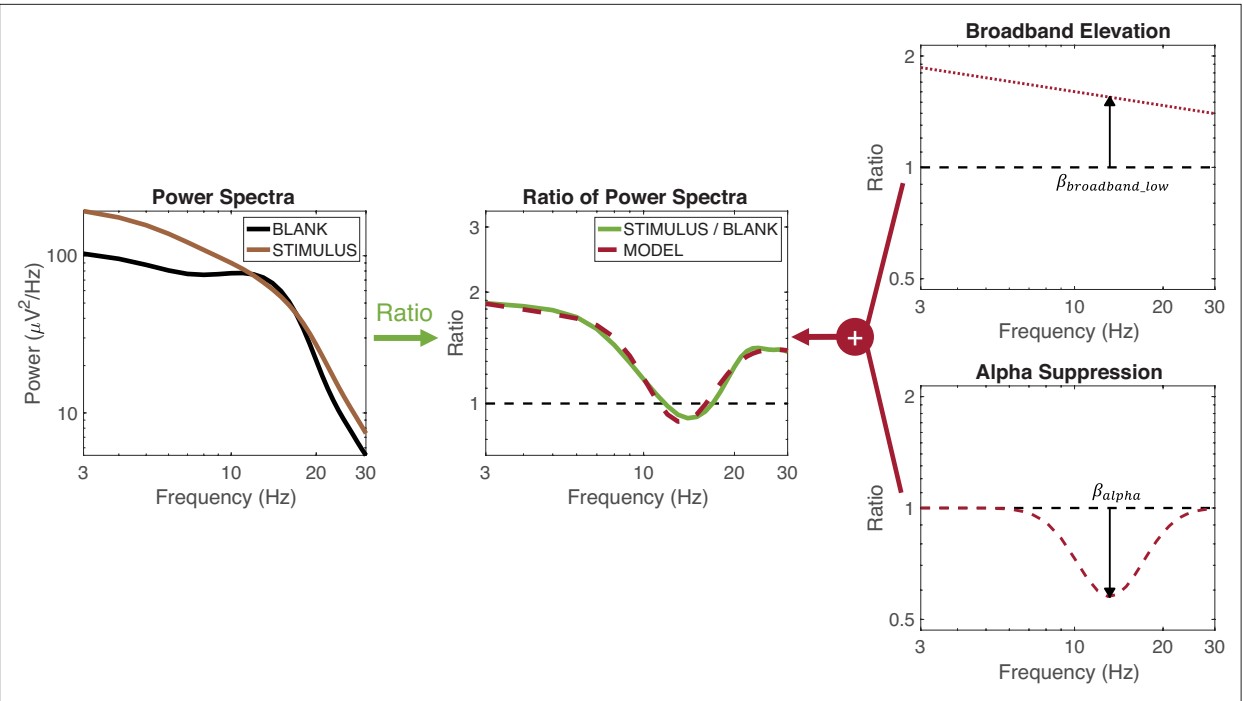

**Figure 11.** Schematics of model-based alpha computation. The left panel shows the average power spectra elicited by the pRF mapping stimuli (brown) and blanks (black) from a representative electrode. The middle panel shows the ratio between the two power spectra (green). This ratio, evaluated on log-log axes, is modeled (dashed red) as the sum of a broadband elevation (right panel, top) and an alpha suppression (right panel, bottom). The magnitude of alpha suppression is the coefficient of the Gaussian bump ($\beta_{alpha}$, black arrow). See *Equation 1* and *makeFigure11.m*.

epoch (see *ecog_prf_regressData.m*). From the time series after removal of the ERP, we computed the PSD during the stimulus presentation period (0–500 ms) in 1 Hz bins. *Welch, 1967* method with a 200 ms length Hann window with 50% overlap to attenuate edge effects. Baseline power spectra were computed for each electrode by taking a geometric average of the power spectra at each frequency bin across blank epochs (see *ecog_prf_spectra.m*).

## pRF analysis

Each participant had multiple identical pRF runs, with the same sequence of 224 trials per run, and 2–6 runs per participant. For each electrode and for each of the 224 trials, we computed the geometric mean of the PSD per frequency bin across the repeated runs, to yield a sequence of 224 PSDs per electrode. We extracted a summary measure of alpha power and broadband power from each of the PSDs, so that for each electrode, we obtained a series of 224 alpha values and 224 broadband values.

### Estimating alpha power from the PSD

We used a model-based approach to compute the alpha responses. This is important because the broadband response extends to low frequencies, and if one just measured the power in a pre-defined range of frequencies corresponding to the alpha band, it would reflect a sum of the two signals rather than the alpha response alone.

To decompose the spectral power into a broadband shift and a change in the alpha oscillation, we modeled the spectral power changes in the 3–26 Hz range as the sum of two responses in the log power /log frequency domain: a linear shift to capture the broadband response and a Gaussian to capture the alpha response (*Figure 11*), according to the formula based on *Hermes et al., 2015* decomposition of broadband and gamma oscillations:

$$\textit{Stimulus evoked change in power} = \textit{Broadband elevation} + \textit{Alpha suppression}$$

$$\log_{10}(\frac{P_s(k)}{P_B(k)}) = (\beta_{broadband\_low} - n(k - \mu)) + \beta_{alpha}G(k|\mu, \alpha),$$

$$k = \log_{10}(frequency),$$

$$G(k|\mu, \sigma) = e^{\frac{-(k-\mu)^2}{2\sigma^2}},$$

(1)

with $8\,Hz < 10^{\mu} < 13\,Hz$.

$P(k)$ is the power spectral density (PSD), either for the stimulus ($P_S$) or the blank ($P_B$). $10^{\mu}$ indicates the peak alpha frequency. We define the change in power as the logarithm of the ratio, so that, for example, a doubling or halving in power are treated symmetrically. Typically, the response to a stimulus is an increase in broadband power and a decrease in the alpha oscillation, so $\beta_{broadband\_low}$ is usually positive and $\beta_{alpha}$ is usually negative. We refer to the broadband component as $\beta_{broadband\_low}$ to distinguish it from the broadband power we estimate at higher frequencies (70–180 Hz). While the two responses might result from a common biological cause, as proposed by *Miller et al., 2009a*, we treat the power elevation at lower and higher frequencies separately here. This ensures that the pRFs estimated from the broadband response (70–180 Hz) and from the alpha responses are independent measures, and that any relationship between them is not an artifact of the decomposition method. Confining the broadband signal to higher frequencies to estimate the broadband pRFs also facilitates comparison with other reports, since most groups exclude low frequencies from their estimates of ECoG broadband data. Hence the term $\beta_{broadband\_low}$ is defined here only to help estimate the alpha response, and is not used directly for fitting pRF models in our main analysis comparing alpha and broadband pRFs.

For two participants, there was a prominent spectral peak just above the alpha band (~24 Hz, beta band), which interfered with this spectral decomposition. In this case, we added a third component to the model, identical to the term for alpha, except with a center constrained to 15–30 Hz instead of 8–13 Hz, and applied the model in the 3–32 Hz range:

$$\log_{10}(\frac{P_S(k)}{P_B(k)}) = (\beta_{broadband\_low} - n(k - \mu)) + \beta_{beta}\, G(k|\mu_2, \sigma_2),$$

$$k = \log_{10}(frequency),$$

$$G(k|\mu_1, \sigma_1) = e^{\frac{-(k-\mu_1)^2}{2\sigma_1^2}}, \quad G(k|\mu_2, \sigma_2) = e^{\frac{-(k-\mu_2)^2}{2\sigma_2^2}},$$

(2)

with $8\,Hz < 10^{\mu_1} < 13\,Hz$ and $15\,Hz < 10^{\mu_2} < 30\,Hz$.

The amplitude of this beta oscillation was modeled as a nuisance variable in that it helped improve our estimate of the alpha oscillation but was not used further in analysis. Including this additional term for the other participants had negligible effect, since there were not clear beta oscillations in other participants, so that the coefficient was usually close to 0. For simplicity, we omitted the beta terms for all datasets except for the two participants who required it. This spectral decomposition was computed for each aperture step in the pRF experiments with the constraint that the peak alpha frequency, $10^{\mu}$, was within ±1 Hz of the peak estimated from averaging across the responses to all apertures for each electrode.

## Estimating broadband power from the PSD

Separately, we estimated the broadband power elevation in the higher frequency range (70–180 Hz) for each electrode and each stimulus location. The broadband power elevation was defined as the ratio of stimulus power ($P_S$) to blank power ($P_B$), where power was computed as the geometric mean across frequencies from 70 to 180 Hz in 1 Hz bins, according to the formula:

$$\textit{Broadband elevation} = \beta_{broadband} = \frac{geometric\ mean\ (P_S(f))}{geometric\ mean\ (P_B(f))},$$

(3)

with $70\,Hz < f < 180\,Hz$ (excluding harmonics of line noise).

The harmonics of power line components were excluded from the averaging: 116–125 and 176–180 Hz from NYU patients and 96–105 and 146–155 Hz from UMCU patients.

## Time courses from pRF analyses

The computations above yield one estimate of alpha power and one estimate of broadband power per trial per electrode, meaning two time series with 224 samples. Visual inspection showed that these two metrics both tended to rise and fall relatively slowly relative to the small changes in position of the sweeping bar aperture. Therefore, the rapid fluctuations in the summary time series were mostly noise. To increase signal-to-noise ratio, each time course was decimated using a low-pass Chebyshev Type I infinite impulse response (IIR) filter of order 3, resulting in 75 time points (see *ecog_prf_fitalpha.m* and *ecog_prf_constructTimeSeries.m*). The same decimation procedure was applied to the binary contrast aperture time series.

## Fitting pRF models

The alpha and broadband power time course and the sequential stimulus contrast apertures were used to estimate two pRFs per electrode (one for alpha, one for broadband), using a Difference of Gaussians (DoG) model (*Zuiderbaan et al., 2012*). The DoG model predicts the response amplitude as the dot product of the binarized stimulus aperture and a pRF. For broadband, the pRF was assumed to be a positive 2D Gaussian (circular) summed with a negative surround. For alpha, the pRF was assumed to be negative with a positive surround. Early analyses indicated a need for the surround, but a lack of precision in estimating its size. For simplicity and to avoid overfitting, we assumed the surround to extend infinitely. Hence, the pRF is a Gaussian with an offset. For each model (alpha and broadband) for each electrode, five parameters were fit: center location of the pRF $(x, y)$, standard deviation of the center Gaussian $(\sigma)$, and the gains of the center $(g_1)$ and the surround $(g_2)$. We estimated these pRF parameters using the analyzePRF toolbox (*Kay et al., 2013a*). We modified the toolbox to incorporate the surround (see *analyzePRFdog.m* in ECoG_utils). The predicted response is obtained by the dot product of aperture and pRF:

$$
\begin{aligned}
RESP &= STIM \cdot PRF \\
&= STIM \cdot \left( g_1 \cdot G_1 \left( x, y, \sigma \right) - g_2 \right).
\end{aligned}
\tag{4}
$$

We optimized the parameters $(x, y, \sigma, g_1,$ and $g_2)$ across all stimulus locations by maximizing the coefficient of determination $(R^2)$ between the 75 predicted time points and observed time series using a least squares non-linear search solver (*lsqcurvefit*), employing the Levenberg-Marquardt algorithm:

$$
R^2 = 1 - \frac{\sum\limits_{stimuli} \left( RESP_{MODEL} - RESP_{DATA} \right)^2}{\sum\limits_{stimuli} RESP_{DATA}{}^2}.
\tag{5}
$$

We set the boundary on the center of Gaussian $(x, y)$ as [−16.6° and 16.6°], two times the maximal extent of the visual stimulus. We performed the fitting twice, once to compute pRF parameters and once to compute model accuracy separately. We use the results of the fit to the complete dataset for each electrode to report pRF parameters. To estimate model accuracy, we fit the data using twofold cross-validation. For cross-validation, each time course was separated into first and second half time courses. Both halves included all horizontal and vertical stimulus positions but reversed in temporal order. We estimated the pRF parameters from each half-time course (37 or 38 time points), and used these parameters to predict the response to the other half of the time course. Then we obtained the cross-validated coefficient of determination as the model accuracy of the concatenated predictions $(R^2)$:

$$
\text{cross-validated } R^2 = 1 - \frac{\sum \left( RESP_{\text{cross-validated MODEL}} - RESP_{DATA} \right)^2}{\sum RESP_{DATA}{}^2}.
\tag{6}
$$

To summarize the data from each electrode, we took the cross-validated variance explained from the split half analysis, and the parameter estimates from the full fit, converting the $x$ and $y$ parameters to polar angle and eccentricity (see *ecog_prf_analyzePRF.m*).

## Comparison of alpha and broadband pRFs

We compared the pRF parameters for the alpha model and broadband models for a subset of electrodes which showed reasonably good fits for each of the two measures. To be included in the comparison, electrodes needed to have a pRF center within the maximal stimulus extent (8.3°) for both broadband and alpha and to have a goodness of fit exceeding thresholds for both broadband and alpha.

The goodness of fit thresholds were defined based on null distributions. The null distributions were computed separately for broadband and alpha. In each case, we shuffled the assignment of pRF parameters to electrodes, computing how accurately the pRF model from one electrode explained the time series from a different electrode. We did this 5000 times to compute a distribution, and chose a variance explained threshold that was higher than 95% of the values in this null distribution. That value was 31% for broadband and 22% for alpha (*Figure 4—figure supplement 1*). Electrodes whose cross-validated variance explained without shuffling exceeded these values were included for the comparison between the two types of signals (assuming their pRF centers were within the 8.3° stimulus aperture).

This procedure resulted in 54 electrodes from four patients. Of these, 35 electrodes had non-zero probability assignments to V1, V2, or V3, 45 to dorsolateral visual areas (TO, LO, V3AB, or IPS), and three to ventral visual areas (hV4, VO, or PHC). The total is greater than 54 because of the probabilistic assignment of electrodes to areas (See *Electrode localization* for the probabilistic assignment). See *Table 2* for details.

To summarize the relationship between broadband and alpha pRF parameters separately for low (V1–V3) vs. high (dorsolateral) visual areas, we used a resampling procedure that accounts for uncertainty of the assignment between electrodes and visual area. We sampled from the 54 electrodes 54 times with replacement, and randomly assigned each electrode to a visual area according to the Wang full probability distributions. After the resampling, on average, 17.7 electrodes were assigned to V1–V3, 23.2 to dorsolateral, and 12.1 to neither. The reason for the 'neither' assignments is that the total probability (100%) of each electrode is the sum of probability of V1-V3, probability of dorsolateral, and probability of elsewhere in the brain. We repeated this procedure 5000 times. We then plotted pRF parameters from the broadband and alpha models as 2D histograms (e.g. *Figure 5*, *Figure 5—figure supplement 1*, and *Figure 6—figure supplement 1*). For the 2D histograms, we computed and plotted 68% confidence intervals derived from the covariance matrix using the function *error_ellipse.m* (https://www.mathworks.com/matlabcentral/fileexchange/4705-error_ellipse).

## Coherence analysis

### Coherence in the high-density grids

We assessed the spatial dependence of coherence in each of the two signals (broadband and alpha) by analyzing data from the two high-density grids (Patients 8 and 9). For each pair of electrodes on a grid, we computed the cross power spectral density (CPSD) in each epoch. The CPSD was computed on the time series after regressing out the ERPs, as described above (*Computing power spectral density*) in 500 ms sliding windows (75% overlap, Hann windows, *Welch, 1967* method). Regressing out the ERPs ensures that coherence measured between electrodes is not a result of the two electrodes having similar stimulus-triggered responses. The CPSD was calculated over the stimulus interval ($-200$–800 ms) in 1 Hz frequency bins (see *ecog_prf_crossspectra.m*). For each electrode pair, and for each of the 224 time series, the magnitude-squared coherence (MSC) was computed and averaged across the $N$ repeated experiments:

$$MSC_{xy}\left(f\right) = \frac{1}{N}\sum_{runs}^{N} \frac{P_{xy}\left(f\right)\ P_{xy}^{*}\left(f\right)}{P_{x}\left(f\right)\ P_{y}\left(f\right)}, \qquad (7)$$

where $P_{xy}$ is the CPSD across two electrodes, and $P_{x}$, $P_{y}$ are the PSDs at each electrode.

This calculation returns an array of time series by frequency bins for each electrode pair (see *ecog_prf_connectivity.m*). We defined the alpha coherence between a seed electrode and all other electrodes on the high-density grid as the coherence at the peak alpha frequency. We defined broadband coherence as the average of the coherence in the broadband frequency range, 70–180 Hz. For every electrode pair on a grid, we then had 224 measures of alpha coherence and 224 measures of

**Table 2.** Overview of electrodes and patients per visual area and figure.

*Electrodes in visual areas (rows 1–3) indicate electrodes with >0% probability of being in an either V1 to V3 or dorsolateral maps, irrespective of the accuracy of population receptive field (pRF) fits. Electrodes with broadband pRFs (rows 4–6) indicate the subset of electrodes from rows 1–3 exceeding a threshold variance explained by the broadband pRF model. Electrodes with broadband and alpha pRFs (rows 7–9) are the subset of electrodes from rows 4–6 satisfying selection criteria both for broadband and alpha pRFs (exceeding threshold variance explained and a pRF center within the stimulus aperture). The totals for rows 'V1–V3 or dorsolateral' are greater than the sum of 'V1-V3' and 'dorsolateral' because some electrodes have greater than 0% probability of being included in both groups (see Electrode Localization, section 4.6). Any electrode with greater than 95% chance of being assigned to no visual area in the atlas was excluded from the table entirely.*

| | Visual area | # of electrodes with probability >0 | Contributing patients (# of electrodes) |
|---|---|---|---|
| Electrodes in visual areas (*Figure 3—figure supplement 1, Figure 4—figure supplement 1*) | V1–V3 | 104 | 1 (1), 2 (15), 3 (6), 4 (13), 5 (6), 8 (54), 9 (9) |
| | Dorsolateral | 303 | 1 (7), 2 (7), 3 (18), 4 (19), 5 (18), 6 (6), 7 (8), 8 (130), 9 (90) |
| | V1–V3 or dorsolateral | 329 | 1 (7), 2 (19), 3 (21), 4 (28), 5 (20), 6 (6), 7 (8), 8 (130), 9 (90) |
| Electrodes with broadband pRFs (*Figures 4 and 7, Figure 4—figure supplement 2, Figure 7—figure supplement 1*) | V1–V3 | 55 | 1 (1), 2 (9), 5 (6), 8 (32), 9 (7) |
| | Dorsolateral | 121 | 1 (1), 2 (2), 3 (1), 4 (1), 5 (4), 7 (1), 8 (63), 9 (48) |
| | V1–V3 or dorsolateral | 131 | 1 (1), 2 (10), 3 (1), 4 (1), 5 (6), 7 (1), 8 (63), 9 (48) |
| Electrodes with broadband and alpha pRFs (*Figures 3, 5 and 6, Figure 1—figure supplement 2, Figure 5—figure supplements 1 and 2, Figure 6—figure supplements 1 and 2*) | V1–V3 | 35 | 2 (6), 5 (5), 8 (20), 9 (4) |
| | Dorsolateral | 45 | 5 (3), 8 (25), 9 (17) |
| | V1–V3 or dorsolateral | 53 | 2 (6), 5 (5), 8 (25), 9 (17) |

broadband coherence. For consistency with the pRF analyses, these time series of alpha and broadband coherence were again decimated into 75 time points.

We then binned the coherence data by electrode pair as a function of inter-electrode distance and by trial as a function of stimulus location. Electrode pairs were binned into inter-electrode distances in multiples of 3 mm. Only electrode pairs for which the seed electrode satisfied the pRF threshold as computed in the previous section were selected (see *Comparison of alpha and broadband pRFs*). We used bootstrapping to compute error bars. For each inter-electrode distance, we randomly sampled electrode pairs with replacement, and averaged the coherence across the electrode pairs and trials. We repeated this procedure 5000 times. The averaged coherences were fitted to the exponential decay function across inter-electrode distances using the least squares method.

## Acknowledgements

We thank Dora Hermes (Mayo Clinic) for help on the model-based method to analyze alpha oscillations, and for feedback on an earlier version of the work; Mike Landy (NYU) for suggesting we compare the spatial spread of attentional cueing with the spatial spread of the alpha pRF; Kaoru Amano (U Tokyo) and his lab for providing feedback on an early version of the work; Ilona Bloem (NYU) for feedback on the manuscript; and Hiromasa Takemura (NIPS, Japan) for providing feedback on the project. This work was supported in part through the NYU IT High Performance Computing resources, services, and staff expertise.

## Additional information

### Competing interests

Iris IA Groen: Reviewing editor, eLife. The other authors declare that no competing interests exist.

### Funding

| Funder | Grant reference number | Author |
|---|---|---|
| National Institute of Mental Health | R01 MH111417 | Nick F Ramsey |
| Japan Society for the Promotion of Science | Overseas Research Fellowship | Kenichi Yuasa |

The funders had no role in study design, data collection and interpretation, or the decision to submit the work for publication.

### Author contributions

Kenichi Yuasa, Conceptualization, Data curation, Software, Formal analysis, Funding acquisition, Validation, Investigation, Visualization, Methodology, Writing – original draft, Project administration, Writing – review and editing; Iris IA Groen, Data curation, Software, Investigation, Methodology, Writing – review and editing; Giovanni Piantoni, Data curation, Software, Investigation; Stephanie Montenegro, Investigation; Adeen Flinker, Sasha Devore, Resources, Investigation, Project administration; Orrin Devinsky, Supervision, Funding acquisition, Investigation, Project administration, Writing – review and editing; Werner Doyle, Resources, Investigation, Methodology; Patricia Dugan, Daniel Friedman, Nick F Ramsey, Resources, Funding acquisition, Investigation, Project administration; Natalia Petridou, Resources, Data curation, Supervision, Funding acquisition, Investigation, Methodology, Project administration, Writing – review and editing; Jonathan Winawer, Conceptualization, Resources, Software, Supervision, Funding acquisition, Investigation, Visualization, Methodology, Project administration, Writing – review and editing

### Author ORCIDs

Kenichi Yuasa https://orcid.org/0000-0002-5697-7537
Iris IA Groen https://orcid.org/0000-0002-5536-6128
Giovanni Piantoni https://orcid.org/0000-0002-5308-926X
Adeen Flinker https://orcid.org/0000-0003-1247-1283

Orrin Devinsky http://orcid.org/0000-0003-0044-4632
Daniel Friedman http://orcid.org/0000-0003-1068-1797
Natalia Petridou http://orcid.org/0000-0002-0783-0387
Jonathan Winawer https://orcid.org/0000-0001-7475-5586

### Ethics

The participants gave informed consent to participate, and the study was approved by the NYU Grossman School of Medicine Institutional Review Board and the ethical committee of the UMCU.

Reviewer #1 (Public Review): https://doi.org/10.7554/eLife.90387.3.sa1
Reviewer #2 (Public Review): https://doi.org/10.7554/eLife.90387.3.sa2
Reviewer #3 (Public Review): https://doi.org/10.7554/eLife.90387.3.sa3
Author response https://doi.org/10.7554/eLife.90387.3.sa4

## Additional files

### Supplementary files

MDAR checklist

### Data availability

The data are shared in BIDS format via Open Neuro (https://doi.org/10.18112/openneuro.ds004194.v3.0.0). The analysis depends on two repositories with MATLAB code: one for conversion to BIDS and pre-processing (ECoG_utils, https://github.com/WinawerLab/ECoG_utils copy archived at *Yuasa, 2025a*), and one for analysis of the pre-processed data (ECoG_alphaPRF, https://github.com/KenYMB/ECoG_alphaPRF copy archived at *Yuasa, 2025b*).

The following dataset was generated:

| Author(s) | Year | Dataset title | Dataset URL | Database and Identifier |
|---|---|---|---|---|
| Groen IIA, Yuasa K, Brands AM, Piantoni G, Montenegro S, Flinker A, Devore S, Devinsky O, Doyle W, Dugan P, Friedman D, Ramsey N, Petridou N, Winawer J | 2025 | Visual ECoG dataset | https://doi.org/10.18112/openneuro.ds004194.v3.0.0 | OpenNeuro, 10.18112/openneuro.ds004194.v3.0.0 |

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

# Appendix 1

### Graphical and statistical support for primary claims

Here, we note which analyses support each major claim in the paper and, where appropriate, how we dealt with nesting (the grouping of electrodes to individual participant).

1. Claim: Alpha responses are accurately predicted by a pRF model (Section 2.3).
   Support:
   - *Figure 4* (left side) shows that for V1 to V3, there is greater variance explained in the alpha responses by the pRF model for visually responsive than for visually non-responsive electrodes (where 'visually responsive' is defined by variance explained by the broadband pRF fits). The difference in variance explained (visually responsive vs non-responsive) is many times larger than the variability, making formal null hypothesis testing superfluous.
   - *Figure 4—figure supplement 2* (left) shows the same pattern for dorsolateral maps.

   Nesting:
   - *Figure 4* (right side) shows that for V1–V3, the pattern observed when pooling across all electrodes is not due to the influence of a single participant's electrodes; the same pattern is observed in individual participants.
   - *Figure 4—figure supplement 2* (right) shows that for dorsolateral electrodes, the pattern for non-visually responsive electrodes is the same in all patients: little to no variance explained by the pRF model. There were only two patients with large numbers of visually responsive electrodes in dorsolateral maps, and in each of these patients the pattern was the same (high variance explained by the alpha pRF model).

2. Claim: Alpha pRFs are larger than broadband pRFs (Section 2.4).
   Support:
   - *Figure 5a* shows the alpha and broadband pRFs for each electrode in V1 to V3 (for electrodes whose probabilistic location is higher for V1 to V3 than for other maps). The alpha pRF is larger in nearly every case (15 out of 17). If we include all electrodes that have non-zero probability of localization in V1 to V3, the pattern is the same (33 out of 35 electrodes have larger size for alpha than for broadband).
   - *Figure 5—figure supplement 1a* shows the same pattern for dorsolateral electrodes (electrodes whose maximum probability is in one of the dorsolateral maps): 34 of 36 electrodes have larger size for alpha than broadband pRFs. If we count all electrodes that have none-zero probability of localization in dorsolateral maps, the numbers are 41 of 45 electrodes.
   - *Figure 5d* (V1 to V3) and *Figure 5—figure supplement 1d* (dorsolateral) plot pRF size vs eccentricity separately for broadband and alpha pRFs. Both of these plots show non-overlapping 68% confidence intervals for broadband vs alpha.
   - *Figure 5c* (V1 to V3) and *Figure 5—figure supplement 1c* (dorsolateral) plot 2D histograms of pRF parameters, alpha vs broadband. The rightmost panel in both plots compares pRF size for alpha vs broadband. Nearly all the values are above the identity line, meaning larger size for alpha pRFs.

   Nesting:
   - In *Figure 5a*, *Figure 5—figure supplement 1a*, showing alpha and broadband pRF solutions for every individual electrode, we grouped and color-coded electrodes by individual patient. The larger size for the alpha pRFs is evident in each individual patient.

3. Claim: The alpha and broadband pRFs have similar locations (Section 2.4).
   Support:
   - *Figure 5a* shows the pRF solutions for broadband and alpha in each electrode (n=17). The broadband pRFs have similar centers and are nearly always inside the alpha pRFs. We report in the text that on average, 92.3% of the broadband pRFs (up to 1-SD) are inside the alpha pRFs (up to 1-SD). To show that this is not a trivial consequence of the alpha pRFs being larger, we recomputed this value after shuffling the electrodes, and the number decreases to 30.0%. We do not think a null hypothesis statistical test is needed to show that these percentages are different.

- The same visualization (*Figure 5—figure supplement 1a*) shows that in dorsolateral maps, the broadband pRFs are inside the alpha pRFs. The calculation shows that the containment percentage is 98.0%, decreasing to 25.7% with a shuffle analysis, again a large effect well above the noise.
- The similarity in pRF polar angle is shown in a 2D histogram in *Figure 5c* (left panel) for V1 to V3. Nearly all the data are concentrated close to the identity line.
- The same pattern is shown for the dorsolateral maps in *Figure 5—figure supplement 1c*.

Nesting:

- The similarity in pRF location between broadband and alpha is observed across individual patients, shown in *Figure 5a*, *Figure 5—figure supplement 1a* (grouped and color coded by patient).

4. Claim: The difference in pRF size is not due to a difference in temporal frequency between broadband and alpha. (Section 2.5)

Support:

- *Figure 6* shows the distributions of broadband pRF sizes, separated into high frequency broadband (70–180 Hz) and low-frequency broadband (3–26 Hz), compared to alpha pRF sizes. The alpha pRF size is more than double those of low- and high-frequency broadband, whereas the low and high frequency broadband pRF sizes are the same as each other.
- *Figure 6—figure supplement 2b* shows 2D histograms of pRF parameters in V1 to V3, comparing low- and high-frequency broadband. The angle, eccentricity, and size histograms all show the majority of the data lying on or near the identity line, confirming that low- and high-frequency broadband pRFs are similar, including the size. This differs from the comparison between high frequency broadband and alpha, in which the pRF sizes differ (see Claim 2 above).

Nesting:

- *Figure 6—figure supplement 2a* shows that pRF sizes for high- and low-frequency broadband for each individual electrode. Electrodes are grouped and color-coded by individual patient. Each patient shows a similar size for low- and high-frequency broadband pRFs.

5. Claim: The accuracy of the alpha pRFs depends on the model-based calculation. (Section 2.6)

Support:

- *Figure 7* compares pRF solutions with and without the model-based baseline correction.
  - Left panel: The figure shows a higher cross-validated variance explained for the model-based method (43% vs 10%, median).
  - Middle panel: The model-based method also shows a consistently negative gain (29 out of 31 electrodes), whereas without correction there is a mixture of positive and negative gain (14 negative, 17 positive), inconsistent with 100 y of measurements of the alpha oscillation.
  - Right panel: The estimated pRF size correlates positively with eccentricity when doing the baseline correction, but not without the correction.

Nesting:

- All three patterns described above are evident in individual patents (*Figure 7—figure supplement 1*).

6. Claim: Alpha oscillations are coherent across a larger spatial extent than broadband signals.

Support:

- *Figure 8* shows the significant differences by bootstrapping between broadband and alpha coherence.

Nesting:

- The results are not averaged but are provided for individual patients.

