## [Editor Report · eLife Assessment]

This intracranial EEG study presents **important** and **convincing** neural evidence supporting the high spatial specificity (receptive field) of visually driven alpha-band oscillation in human brains and its potential role in exogenous cuing attention. The work challenges the predominant view about the role of alpha-band oscillation in visual attention and advocates that stimulus-driven alpha suppression is precisely tuned and might contribute to exogenous spatial attention.

---

## [Referee Report · Reviewer #1 (Public Review)]

In this study, the authors build upon previous research that utilized non-invasive EEG and MEG by analyzing intracranial human ECoG data with high spatial resolution. They employed a receptive field mapping task to infer the retinotopic organization of the human visual system. The results present compelling evidence that the spatial distribution of human alpha oscillations is highly specific and functionally relevant, as it provides information about the position of a stimulus within the visual field.

Using state-of-the-art modeling approaches, the authors not only strengthen the existing evidence for the spatial specificity of the human dominant rhythm but also provide new quantification of its functional utility, specifically in terms of the size of the receptive field relative to the one estimated based on broad band activity.

---

## [Referee Report · Reviewer #2 (Public Review)]

Summary:

In this work, Yuasa et al. aimed to study the spatial resolution of modulations in alpha frequency oscillations (~10Hz) within the human occipital lobe. Specifically, the authors examined the receptive field (RF) tuning properties of alpha oscillations, using retinotopic mapping and invasive electroencephalogram (iEEG) recordings. The authors employ established approaches for population RF mapping, together with a careful approach to isolating and dissociating overlapping, but distinct, activities in the frequency domain. Whereby, the authors dissociate genuine changes in alpha oscillation amplitude from other superimposed changes occurring over a broadband range of the power spectrum. Together, the authors used this approach to test how spatially tuned estimated RFs were when based on alpha range activity, vs. broadband activities (focused on 70-180Hz). Consistent with a large body of work, the authors report clear evidence of spatially precise RFs based on changes in alpha range activity. However, the size of these RFs were far larger than those reliably estimated using broadband range activity at the same recording site. Overall, the work reflects a rigorous approach to a previously examined question, for which improved characterization leads to improved consistency in findings and some advance of prior work.

Strengths:

Overall, the authors take a careful and well-motivated approach to data analyses. The authors successfully test a clear question with a rigorous approach and provide strong supportive findings. Firstly, well-established methods are used for modeling population RFs. Secondly, the authors employ contemporary methods for dissociating unique changes in alpha power from superimposed and concomitant broadband frequency range changes. This is an important confound in estimating changes in alpha power not employed in prior studies. The authors show this approach produces more consistent and robust findings than standard band-filtering approaches. As noted below, this approach may also account for more subtle differences when compared to prior work studying similar effects.

Original Weaknesses:

- Theoretical framing: The authors frame their study as testing between two alternative views on the organization, and putative functions, of occipital alpha oscillations: (i) alpha oscillation amplitude reflects broad shifts in arousal state, with large spatial coherence and uniformity across cortex; (ii) alpha oscillation amplitude reflects more specific perceptual processes and can be modulated at local spatial scales. However, in the introduction this framing seems mostly focused on comparing some of the first observations of alpha with more contemporary observations. Therefore, I read their introduction to more reflect the progress in studying alpha oscillations from Berger's initial observations to the present. I am not aware of a modern alternative in the literature that posits alpha to lack spatially specific modulations. I also note this framing isn't particularly returned to in the discussion. A second important variable here is the spatial scale of measurement. It follows that EEG based studies will capture changes in alpha activity up to the limits of spatial resolution of the method (i.e. limited in ability to map RFs). This methodological distinction isn't as clearly mentioned in the introduction, but is part of the author's motivation. Finally, as noted below, there are several studies in the literature specifically addressing the authors question, but they are not discussed in the introduction.

- Prior studies: There are important findings in the literature preceding the author's work that are not sufficiently highlighted or cited. In general terms, the spatio-temporal properties of the EEG/iEEG spectrum are well known (i.e. that changes in high frequency activity are more focal than changes in lower frequencies). Therefore, the observations of spatially larger RFs for alpha activities is highly predicted. Specifically, prior work has examined the impact of using different frequency ranges to estimate RF properties, for example ECoG studies in the macaque by Takura et al. NeuroImage (2016) [PubMed: 26363347], as well as prior ECoG work by the author's team of collaborators (Harvey et al., NeuroImage (2013) [PubMed: 23085107]), as well as more recent findings from other groups (Luo et al., (2022) BioRxiv: https://doi.org/10.1101/2022.08.28.505627). Also, a related literature exists for invasively examining RF mapping in the time-voltage domain, which provides some insight into the author's findings (as this signal will be dominated by low-frequency effects). The authors should provide a more modern framing of our current understanding of the spatial organization of the EEG/iEEG spectrum, including prior studies examining these properties within the context of visual cortex and RF mapping. Finally, I do note that the author's approach to these questions do reflect an important test of prior findings, via an improved approach to RF characterization and iEEG frequency isolation, which suggests some important differences with prior work.

- Statistical testing: The authors employ many important controls in their processing of data. However, for many results there is only a qualitative description or summary metric. It appears very little statistical testing was performed to establish reported differences. Related to this point, the iEEG data is highly nested, with multiple electrodes (observations) coming from each subject, how was this nesting addressed to avoid bias?

[Editors' note: the authors have addressed the original concerns.]

---

## [Referee Report · Reviewer #3 (Public Review)]

Summary:

This study tackles the important subject of sensory driven suppression of alpha oscillations using a unique intracranial dataset in human patients. Using a model-based approach to separate changes in alpha oscillations from broadband power changes, the authors try to demonstrate that alpha suppression is spatially tuned, with similar center location as high broadband power changes, but much larger receptive field. They also point to interesting differences between low-order (V1-V3) and higher-order (dorsolateral) visual cortex. While I find some of the methodology convincing, I also find significant parts of the data analysis, statistics and their presentation incomplete. Thus, I find that some of the main claims are not sufficiently supported. If these aspects could be improved upon, this study could potentially serve as an important contribution to the literature with implications for invasive and non-invasive electrophysiological studies in humans.

Strengths:

The study utilizes a unique dataset (ECOG & high-density ECOG) to elucidate an important phenomenon of visually driven alpha suppression. The central question is important and the general approach is sound. The manuscript is clearly written and the methods are generally described transparently (and with reference to the corresponding code used to generate them). The model-based approach for separating alpha from broadband power changes is especially convincing and well-motivated. The link to exogenous attention behavioral findings (figure 8) is also very interesting. Overall, the main claims are potentially important, but they need to be further substantiated (see weaknesses).

Original Weaknesses:

I have three major concerns:

(1) Low N / no single subject results/statistics: The crucial results of Figure 4,5 hang on 53 electrodes from four patients (Table 2). Almost half of these electrodes (25/53) are from a single subject. Data and statistical analysis seem to just pool all electrodes, as if these were statistically independent, and without taking into account subject-specific variability. The mean effect per each patient was not described in text or presented in figures. Therefore, it is impossible to know if the results could be skewed by a single unrepresentative patient. This is crucial for readers to be able to assess the robustness of the results. N of subjects should also be explicitly specified next to each result.

(2) Separation between V1-V3 and dorsolateral electrodes: Out of 53 electrodes, 27 were doubly assigned as both V1-V3 and dorsolateral (Table 2, Figures 4,5). That means that out of 35 V1-V3 electrodes, 27 might actually be dorsolateral. This problem is exasperated by the low N. for example all the 20 electrodes in patient 8 assigned as V1-V3 might as well be dorsolateral. This double assignment didn't make sense to me and I wasn't convinced by the authors' reasoning. I think it needlessly inflates the N for comparing the two groups and casts doubts on the robustness of these analyses.

(3) Alpha pRFs are larger than broadband pRFs: first, as broadband pRF models were on average better fit to the data than alpha pRF models (dark bars in Supp Fig 3. Top row), I wonder if this could entirely explain the larger Alpha pRF (i.e. worse fits lead to larger pRFs). There was no anlaysis to rule out this possibility. Second, examining closely the entire 2.4 section there wasn't any formal statistical test to back up any of the claims (not a single p-value is mentioned). It is crucial in my opinion to support each of the main claims of the paper with formal statistical testing.

[Editors' note: the authors have addressed the original concerns.]

---

## [Author Response]

The following is the authors’ response to the original reviews

**Reviewer #1 (Public Review):**
SummaryIn this study, the authors build upon previous research that utilized non-invasive EEG and MEG by analyzing intracranial human ECoG data with high spatial resolution. They employed a receptive field mapping task to infer the retinotopic organization of the human visual system. The results present compelling evidence that the spatial distribution of human alpha oscillations is highly specific and functionally relevant, as it provides information about the position of a stimulus within the visual field.

Using state-of-the-art modeling approaches, the authors not only strengthen the existing evidence for the spatial specificity of the human dominant rhythm but also provide new quantification of its functional utility, specifically in terms of the size of the receptive field relative to the one estimated based on broad band activity.

We thank the reviewer for their positive summary.

Weakness 1.1The present manuscript currently omits the complementary view that the retinotopic map of the visual system might be related to eye movement control. Previous research in non-human primates using microelectrode stimulation has clearly shown that neuronal circuits in the visual system possess motor properties (e.g. Schiller and Styker 1972, Schiller and Tehovnik 2001). More recent work utilizing Utah arrays, receptive field mapping, and electrical stimulation further supports this perspective, demonstrating that the retinotopic map functions as a motor map. In other words, neurons within a specific area responding to a particular stimulus location also trigger eye movements towards that location when electrically stimulated (e.g. Chen et al. 2020).

Similarly, recent studies in humans have established a link between the retinotopic variation of human alpha oscillations and eye movements (e.g., Quax et al. 2019, Popov et al. 2021, Celli et al. 2022, Liu et al. 2023, Popov et al. 2023). Therefore, it would be valuable to discuss and acknowledge this complementary perspective on the functional relevance of the presented evidence in the discussion section.

The reviewer notes that we do not discuss the oculomotor system and alpha oscillations. We agree that the literature relating eye movements and alpha oscillations are relevant.

At the Reviewer’s suggestion, we added a paragraph on this topic to the first section of the Discussion (section 3.1, “Other studies have proposed … “).

**Reviewer #2 (Public Review):**
Summary:In this work, Yuasa et al. aimed to study the spatial resolution of modulations in alpha frequency oscillations (~10Hz) within the human occipital lobe. Specifically, the authors examined the receptive field (RF) tuning properties of alpha oscillations, using retinotopic mapping and invasive electroencephalogram (iEEG) recordings. The authors employ established approaches for population RF mapping, together with a careful approach to isolating and dissociating overlapping, but distinct, activities in the frequency domain. Whereby, the authors dissociate genuine changes in alpha oscillation amplitude from other superimposed changes occurring over a broadband range of the power spectrum. Together, the authors used this approach to test how spatially tuned estimated RFs were when based on alpha range activity, vs. broadband activities (focused on 70-180Hz). Consistent with a large body of work, the authors report clear evidence of spatially precise RFs based on changes in alpha range activity. However, the size of these RFs were far larger than those reliably estimated using broadband range activity at the same recording site. Overall, the work reflects a rigorous approach to a previously examined question, for which improved characterization leads to improved consistency in findings and some advance of prior work.

We thank the reviewer for the summary.

Strengths:Overall, the authors take a careful and well-motivated approach to data analyses. The authors successfully test a clear question with a rigorous approach and provide strong supportive findings. Firstly, well-established methods are used for modeling population RFs. Secondly, the authors employ contemporary methods for dissociating unique changes in alpha power from superimposed and concomitant broadband frequency range changes. This is an important confound in estimating changes in alpha power not employed in prior studies. The authors show this approach produces more consistent and robust findings than standard band-filtering approaches. As noted below, this approach may also account for more subtle differences when compared to prior work studying similar effects.

We thank the reviewer for the positive comments.

Weaknesses:Weakness 2.1 Theoretical framing:The authors frame their study as testing between two alternative views on the organization, and putative functions, of occipital alpha oscillations: (i) alpha oscillation amplitude reflects broad shifts in arousal state, with large spatial coherence and uniformity across cortex; (ii) alpha oscillation amplitude reflects more specific perceptual processes and can be modulated at local spatial scales. However, in the introduction this framing seems mostly focused on comparing some of the first observations of alpha with more contemporary observations. Therefore, I read their introduction to more reflect the progress in studying alpha oscillations from Berger's initial observations to the present. I am not aware of a modern alternative in the literature that posits alpha to lack spatially specific modulations. I also note this framing isn't particularly returned to in the discussion.

This was helpful feedback. We have rewritten nearly the entire Introduction to frame the study differently. The emphasis is now on the fact that several intracranial studies of spatial tuning of alpha (in both human and macaque) tend to show *increases* in alpha due to visual stimulation, in contrast to a century of MEG/EEG studies, from Berger to the present, showing *decreases*. We believe that the discrepancy is due to an interaction between measurement type and brain signals. Specifically, intracranial measurements sum decreases in alpha oscillations and increases in broadband power on the same trials, and both signals can be large. In contrast, extracranial measures are less sensitive to the broadband signals and mostly just measure the alpha oscillation. Our study reconciles this discrepancy by removing the baseline broadband power increases, thereby isolating the alpha oscillation, and showing that with iEEG spatial analyses, the alpha oscillation decreases with visual stimulation, consistent with EEG and MEG results.

Weakness 2.2 A second important variable here is the spatial scale of measurement.It follows that EEG based studies will capture changes in alpha activity up to the limits of spatial resolution of the method (i.e. limited in ability to map RFs). This methodological distinction isn't as clearly mentioned in the introduction, but is part of the author's motivation. Finally, as noted below, there are several studies in the literature specifically addressing the authors question, but they are not discussed in the introduction.

The new Introduction now explicitly contrasts EEG/MEG with intracranial studies and refers to the studies below.

Weakness 2.3 Prior studies:There are important findings in the literature preceding the author's work that are not sufficiently highlighted or cited. In general terms, the spatio-temporal properties of the EEG/iEEG spectrum are well known (i.e. that changes in high frequency activity are more focal than changes in lower frequencies). Therefore, the observations of spatially larger RFs for alpha activities is highly predicted. Specifically, prior work has examined the impact of using different frequency ranges to estimate RF properties, for example ECoG studies in the macaque by Takura et al. NeuroImage (2016) [PubMed: 26363347], as well as prior ECoG work by the author's team of collaborators (Harvey et al., NeuroImage (2013) [PubMed: 23085107]), as well as more recent findings from other groups (Luo et al., (2022) BioRxiv: https://doi.org/10.1101/2022.08.28.505627). Also, a related literature exists for invasively examining RF mapping in the time-voltage domain, which provides some insight into the author's findings (as this signal will be dominated by low-frequency effects). The authors should provide a more modern framing of our current understanding of the spatial organization of the EEG/iEEG spectrum, including prior studies examining these properties within the context of visual cortex and RF mapping. Finally, I do note that the author's approach to these questions do reflect an important test of prior findings, via an improved approach to RF characterization and iEEG frequency isolation, which suggests some important differences with prior work.

Thank you for these references and suggestions. Some of the references were already included, and the others have been added.

There is one issue where we disagree with the Reviewer, namely that “the observations of spatially larger RFs for alpha activities is highly predicted”. We agree that alpha oscillations and other *low frequency rhythms* tend to be less focal than high frequency responses, but there are also low frequency non-rhythmic signals, and these can be spatially focal. We show this by demonstrating that pRFs solved using low frequency responses outside the alpha band (both below and above the alpha frequency) are small, similar to high frequency broadband pRFs, but differing from the large pRFs associated with alpha oscillations. Hence we believe the degree to which signals are focal is more related to the degree of rhythmicity than to the temporal frequency *per se*. While some of these results were already in the supplement, we now address the issue more directly in the main text in a new section called, “2.5 The difference in pRF size is not due to a difference in temporal frequency.”

We incorporated additional references into the Introduction, added a new section on low frequency broadband responses to the Results (section 2.5), and expanded the Discussion (section 3.2) to address these new references.

Weakness 2.4 Statistical testing:The authors employ many important controls in their processing of data. However, for many results there is only a qualitative description or summary metric. It appears very little statistical testing was performed to establish reported differences. Related to this point, the iEEG data is highly nested, with multiple electrodes (observations) coming from each subject, how was this nesting addressed to avoid bias?

We reviewed the primary claims made in the manuscript and for each claim, we specify the supporting analyses and, where appropriate, how we address the issue of nesting. Although some of these analyses were already in the manuscript, many of them are new, including all of the analyses concerning nesting. We believe that putting this information in one place will be useful to the reader, and we now include this text as a new section in supplement, Graphical and statistical support for primary claims.

**Reviewer #2 (Recommendations For The Authors):**
Recommendation 2.1:Data presentation: In several places, the authors discuss important features of cortical responses as measured with iEEG that need to be carefully considered. This is totally appropriate and a strength of the author's work, however, I feel the reader would benefit from more depiction of the time-domain responses, to help better understand the authors frequency domain approach. For example, Figure 1 would benefit from showing some form of voltage trace (ERP) and spectrogram, not just the power spectra. In addition, part (a) of Figure 1 could convey some basic information about the timing of the experimental paradigm.

We changed panel A of Figure 1 to include the timing of the experimental paradigm, and we added panels C and D to show the electrode time series before and after regression out of the ERP.

Recommendation 2.2Update introduction to include references to prior EEG/iEEG work on spatial distribution across frequency spectrum, and importantly, prior work mapping RFs with different frequencies.

We have addressed this issue and re-written our introduction. Please refer to our response in Public Review for further details.

Recommendation 2.3Figure 3 has several panels and should be labeled to make it easier to follow.The dashed line in lower power spectra isn't defined in a legend and is missing from the upper panel - please clarify.

We updated Figure 3 and reordered the panels to clarify how we computed the summary metrics in broadband and alpha for each stimulus location (i.e., the “ratio” values plotted in panel B). We also simplified the plot of the alpha power spectrum. It now shows a dashed line representing a baseline-corrected response to the mapping stimulus, which is defined in the legend and explained in the caption.

Recommendation 2.4Power spectra are always shown without error shading, but they are mean estimates.

We added error shading to Figures 1, 2 and 3.

Recommendation 2.5The authors deal with voltage transients in response to visual stimulation, by subtracting out the trail averaged mean (commonly performed). However, the efficacy of this approach depends on signal quality and so some form of depiction for this processing step is needed.

We added a depiction of the processing steps for regressing out the averaged responses in Figure 1 in an example electrode (panels C and D). We also show in the supplement the effect of regressing out the ERP on all the electrode pRFs. We have added Supplementary Figure 1-2.

Recommendation 2.6I have a similar request for the authors latency correction of their data, where they identified a timing error and re-aligned the data without ground truth. Again, this is appropriate, but some depiction of the success of this correction is very critical for confirming the integrity of the data.

We now report more detail on the latency correction, and also point out that any small error in the estimate would not affect our conclusions (4.6 ECoG data analysis | Data epoching). The correction was important for a prior paper on temporal dynamics (Groen et al, 2022), which used data from the same participants and estimated the latency of responses. In this paper, our analyses are in the spectral domain (and discard phase), so small temporal shifts are not critical. We now also link to the public code associated with that paper, which implemented the adjustment and quantified the uncertainty in the latency adjustment.

More details on latency adjustment provided in section 4.6.

Recommendation 2.7In many places the authors report their data shows a 'summary' value, please clarify if this means averaging or summation over a range.

For both broadband and alpha, we derive one summary value (a scalar) for trial for each stimulus. For broadband, the summary metric is the ratio of power during a given trial and power during blanks, where power in a trial is the geometric mean of the power at each frequency within the defined band. This is equation 3 in the methods, which is now referred to the first time that summary metrics are mentioned in the results. For alpha, the summary metric is the height of the Gaussian from our model-based approach. This is in equations 1 and 2, and is also now referred to the first time summary metrics are mentioned in the results.

We added explanation of the summary metrics in the figure captions and results where they are first used, and also referred to the equations in the methods where they are defined.

Recommendation 2.8The authors conclude: "we have discovered that spectral power changes in the alpha range reflect both suppression of alpha oscillations and elevation of broadband power." It might not have been the intention, but 'discovered' seems overstated.

We agree and changed this sentence.

Recommendation 2.9Supp Fig 9 is a great effort by the authors to convey their findings to the reader, it should be a main figure.

We are glad you found Supplementary Figure 9 valuable. We moved this figure to the main text.

**Reviewer #3 (Public Review):**
Summary:This study tackles the important subject of sensory driven suppression of alpha oscillations using a unique intracranial dataset in human patients. Using a model-based approach to separate changes in alpha oscillations from broadband power changes, the authors try to demonstrate that alpha suppression is spatially tuned, with similar center location as high broadband power changes, but much larger receptive field. They also point to interesting differences between low-order (V1-V3) and higher-order (dorsolateral) visual cortex. While I find some of the methodology convincing, I also find significant parts of the data analysis, statistics and their presentation incomplete. Thus, I find that some of the main claims are not sufficiently supported. If these aspects could be improved upon, this study could potentially serve as an important contribution to the literature with implications for invasive and non-invasive electrophysiological studies in humans.

We thank the reviewer for the summary.

Strengths:The study utilizes a unique dataset (ECOG & high-density ECOG) to elucidate an important phenomenon of visually driven alpha suppression. The central question is important and the general approach is sound. The manuscript is clearly written and the methods are generally described transparently (and with reference to the corresponding code used to generate them). The model-based approach for separating alpha from broadband power changes is especially convincing and well-motivated. The link to exogenous attention behavioral findings (figure 8) is also very interesting. Overall, the main claims are potentially important, but they need to be further substantiated (see weaknesses).

We thank the reviewer for the positive comments.

Weaknesses:I have three major concerns:Weakness 3.1. Low N / no single subject results/statistics:The crucial results of Figure 4,5 hang on 53 electrodes from four patients (Table 2). Almost half of these electrodes (25/53) are from a single subject. Data and statistical analysis seem to just pool all electrodes, as if these were statistically independent, and without taking into account subject-specific variability. The mean effect per each patient was not described in text or presented in figures. Therefore, it is impossible to know if the results could be skewed by a single unrepresentative patient. This is crucial for readers to be able to assess the robustness of the results. N of subjects should also be explicitly specified next to each result.

We have added substantial changes to deal with subject specific effects, including new results and new figures.

• Figure 4 now shows variance explained by the alpha pRF broken down by each participant for electrodes in V1 to V3. We also now show a similar figure for dorsolateral electrodes in Supplementary Figure 4-2.

• Figure 5, which shows results from individual electrodes in V1 to V3, now includes color coding of electrodes by participant to make it clear how the electrodes group with participant. Similarly, for dorsolateral electrodes, we show electrodes grouped by participant in Supplementary Figure 5-1. Same for Supplementary Figure 6-2.

• Supplementary Figure 7-2 now shows the benefits of our model-based approach for estimating alpha broken down by individual participants.

• We also now include a new section in the supplement that summarizes for every major claim, what the supporting data are and how we addressed the issue of nesting electrodes by participant, section Graphical and statistical support for primary claims.

Weakness 3.2. Separation between V1-V3 and dorsolateral electrodes:Out of 53 electrodes, 27 were doubly assigned as both V1-V3 and dorsolateral (Table 2, Figures 4,5). That means that out of 35 V1-V3 electrodes, 27 might actually be dorsolateral. This problem is exasperated by the low N. for example all the 20 electrodes in patient 8 assigned as V1-V3 might as well be dorsolateral. This double assignment didn't make sense to me and I wasn't convinced by the authors' reasoning. I think it needlessly inflates the N for comparing the two groups and casts doubts on the robustness of these analyses.

Electrode assignment was probabilistic to reflect uncertainty in the mapping between location and retinotopic map. The probabilistic assignment is handled in two ways.

(1) For visualizing results of single electrodes, we simply go with the maximum probability, so no electrode is visualized for both groups of data. For example, Figure 5a (V1-V3) and supplementary Figure 5-1a (dorsolateral electrodes) have no electrodes in common: no electrode is in both plots.

(2) For quantitative summaries, we sample the electrodes probabilistically (for example Figures 4, 5c). So, if for example, an electrode has a 20% chance of being in V1 to V3, and 30% chance of being in dorsolateral maps, and a 50% chance of being in neither, the data from that electrode is used in only 20% of V1-V3 calculations and 30% of dorsolateral calculations. In 50% of calculations, it is not used at all. This process ensures that an electrode with uncertain assignment makes no more contribution to the results than an electrode with certain assignment. An electrode with a low probability of being in, say, V1-V3, makes little contribution to any reported results about V1-V3. This procedure is essentially a weighted mean, which the reviewer suggests in the recommendations. Thus, we believe there is not a problem of “double counting”.

The alternative would have been to use maximum probability for all calculations. However, we think that doing so would be misleading, since it would not take into account uncertainty of assignment, and would thus overstate differences in results between the maps.

We now clarify in the Results that for probabilistic calculations, the contribution of an electrode is limited by the likelihood of assignment (Section 2.3). We also now explain in the methods why we think probabilistic sampling is important.

Weakness 3.3. Alpha pRFs are larger than broadband pRFs:First, as broadband pRF models were on average better fit to the data than alpha pRF models (dark bars in Supp Fig 3. Top row), I wonder if this could entirely explain the larger Alpha pRF (i.e. worse fits lead to larger pRFs). There was no anlaysis to rule out this possibility.

We addressed this question in a new paragraph in Discussion section 3.1 (“What is the function of the large alpha pRFs?”, paragraph beginning… “Another possible interpretation is that the poorer model fit in the alpha pRF is due to lower signal-to-noise”). This paragraph both refers to prior work on the relationship between noise and pRF size and to our own control analyses (Supplementary Figure 5-2).

Weakness 3.4 StatisticsSecond, examining closely the entire 2.4 section there wasn't any formal statistical test to back up any of the claims (not a single p-value is mentioned). It is crucial in my opinion to support each of the main claims of the paper with formal statistical testing.

We agree that it is important for the reader to be able to link specific results and analyses to specific claims. We are not convinced that null hypothesis statistical testing is always the best approach. This is a topic of active debate in the scientific community.

We added a new section that concisely states each major claim and explicitly annotates the supporting evidence. (Section 4.7). Please also refer to our responses to Reviewer #2 regarding statistical testing (Reviewer weakness 2.4 “Statistical testing”)

Weakness 3.5 SummaryWhile I judge these issues as crucial, I can also appreciate the considerable effort and thoughtfulness that went into this study. I think that addressing these concerns will substantially raise the confidence of the readership in the study's findings, which are potentially important and interesting.

We again thank the reviewer for the positive comments.

**Reviewer #3 (Recommendations For The Authors):**
Suggestions for how to address the three major concerns:Suggestion 3.1.I am very well aware that it's very hard to have n=30 in a visual cortex ECOG study. That's fine. Best practice would be to have a linear mixed effects model with patients as a random effect. However, for some figures with just 3-4 patients (Figure 4,5) the sample size might be too small even for that. At the very minimum, I would expect to show in figures/describe in text all results per patient (perhaps one can do statistics within each patient, and show for each patient that the effect is significant). Even in primate studies with just two subjects it is expected to show that the results replicate for subject A and B. It is necessary to show that your results don't depend on a single unrepresentative subject. And if they do, at least be transparent about it.

We have addressed this thoroughly. Please see response to Weakness 3.1 (“Low N / no single subject results/statistics”).

Suggestion 3.2.I just don't get it. I would simply assign an electrode to V1-V3 or dorsolateral cortex based on which area has the highest probability. It doesn't make sense to me that an electrode that has 60% of being in dorsolateral cortex and only 10% to be in V1-V3 would be assigned as both V1-V3 and dorsolateral. Also, what's the rationale to include such electrode in the analysis for let's say V1-V3 (we have weak evidence to believe it's there)? I would either assign electrodes based on the highest probability, or alternatively do a weighted mean based on the probability of each electrode belonging to each region group (e.g. electrode with 40% to be in V1-V3, will get twice the weight as an electrode who has 20% to be in V1-V3) but this is more complicated.

We have addressed this issue. Please refer to our response in Public Review (“Weakness 3.2 Separation between V1-V3 and dorsolateral”) for details.

Suggestion 3.3.First, to exclude the possibility that alpha pRF are larger simply because they have a worse fit to the neural data, I would show if there is a correlation between the goodnessof-fit and pRF size (for alpha and broadband signals, separately). No [negative] correlation between goodness-of-fit and pRF size would be a good sign. I would also compare alpha & broadband receptive field size when controlling for the goodness-of-fit (selecting electrodes with similar goodness-of-fit for both signals). If the results replicate this way it would be convincing.Second, there are no statistical tests in section 2.4, possibly also in others. Even if you employ bootstrap / Monte-Carlo resampling methods you can extract a p-value.

We have addressed this issue. Please refer to our response in Public Review Point 3.3 (“Alpha pRFs are larger than broadband pRFs”) for further details.

Suggestion 3.4.Also, I don't understand the resampling procedure described in lines 652-660: "17.7 electrodes were assigned to V1-V3, 23.2 to dorsolateral, and 53 to either " - but 17.7 + 23.2 doesn't add up to 53. It also seems as if you assign visual areas differently in this resampling procedure than in the real data - "and randomly assigned each electrode to a visual area according to the Wang full probability distributions". If you assign in your actual data 27 electrodes to both visual areas, the same should be done in the resampling procedure (I would expect exactly 35 V1-V3 and 45 dorsolateral electrodes in every resampling, just the pRFs will be shuffled across electrodes).

We apologize for the confusion.

We fixed the sentence above, clarified the caption to Table 2, and also explained the overall strategy of probabilistic resampling better. See response to Public Review point 3.2 for details.

Suggestion 3.5.These are rather technical comments but I believe they are crucial points to address in order to support your claims. I genuinely think your results are potentially interesting and important but these issues need to be first addressed in a revision. I also think your study may carry implications beyond just the visual domain, as alpha suppression is observed for different sensory modalities and cortical regions. Might be useful to discuss this in the discussion section.

Agree. We added a paragraph on this point to the Discussion (very end of 3.2).